# Bulk-cusp microstructure for controllable multi-directional liquid spreading

Songjie Dai ⬭, Hui Zhang ⬭ ✉, Yang Liu, Fenghao Yi, Kaibin Shi & Guangneng Dong ⬭

Controllable wetting of liquid on solid surface is meaningful for advanced science and engineering. Current researches about controllable liquid spreading are generally limited to unidirectional modes, while achieving multi-directional spreading on microstructured surfaces remains challenging. Herein, we propose a novel type of bulk-cusp microstructure, exhibiting 0 to 4-directional spreading of droplet without external energy input. This behavior occurs not only under single-drop deposition but also under continuous liquid injection. The bulk structure is assumed to have cross or square shape, implying relatively high and low coverage ratio of the precursor film, respectively. Owing to the drag effect of the precursor film, the cross-cusp microstructure facilitates controllable spreading of the main droplet, whereas the square-cusp microstructure just has guidance action on precursor film due to its low coverage ratio. Mechanism analysis reveals the capillary forces generated from the narrow gaps between adjacent cusps effectively separate the precursor film. The area coverage ratio of precursor film determined by the shape of the bulk influences the coupling or decoupling of the droplet body and precursor film. Such controllable multi-directional liquid spreading enables applications in lubrication enhancement and smart evaporation cooling.

Controlling liquid wettability on solid surfaces is of great significance in interfacial science and engineering, with broad applications in microfluidic chips[1–3], water harvesting[4–6], thermal management[7–9], biochemical analysis[10–12], and mechanical lubrication[13–15]. Natural systems have evolved intricate surface microstructures to enable directional liquid spreading, leveraging wettability gradients and microstructures. The peristome of *Nepenthes alata*, for instance, facilitates unidirectional liquid spreading via hierarchical microstructures[16–18], cactus spines utilize asymmetric wedge-shaped wettability gradient to drive water spreading and collection[19–21], *Araucaria* leaves guide liquids of different surface tensions by three-dimensional Laplace pressure gradients[22], while the asymmetric concave structures on *Crassula muscosa* leverage local curvature gradients to enable the selective spreading of the same liquid[23]. These biological strategies reveal the significant effect of wettability gradients and microstructures on liquid spreading, providing inspiration for the design of advanced artificial surfaces. Wettability gradients propel liquids spreading from low to high surface energy regions, but they often suffer from limited uniformity and unstable spreading performance[24–26]. In contrast, capillary action induced by microstructures enables stable long-lasting directional spreading of liquid through confined geometries[27–29]. In available literature, most artificial designs remain confined to unidirectional spreading, and only limited cases of selective single-direction transport have been reported.

In recent years, artificial surfaces have been developed to achieve multi-directional spreading through complex three-dimensional geometries. For example, the three-dimensional asymmetric Fang-like structures regulate liquid spreading modes through multiple curvature gradients[30], whereas capillary ratchet and curvature ratchet surfaces achieve directional liquid spreading by exploiting asymmetric

Key Laboratory of Education Ministry for Modern Design & Rotor-Bearing System, Xi'an Jiaotong University, Xi'an, China. ✉e-mail: zhanghui7@xjtu.edu.cn

Laplace pressure distribution[31,32]. Additionally, designs such as multi-asymmetric magnetized surfaces and reconfigurable encoded recti-fiers utilize external magnetic fields to control liquid transport paths, enhancing the flexibility of directional regulation[33–35]. Nevertheless, most existing approaches depend on complex three-dimensional architectures such as inclined micropillars, re-entrant cavities, or reconfigurable patterns. These designs, while conceptually valuable, suffer from fabrication complexity, limited scalability, and strong dependence on external fields. Previous studies also focused on guiding different liquids along preset pathways, but rarely demon-strated how a single liquid can be programmed into multiple spreading modes within one type of microstructure. As most methods require external fields, pump-free multi-directional spreading remains uncommon. Therefore, current strategies often meet only part of the requirements for controllable liquid transport, and there remains a clear need for energy-free, scalable, and versatile strategies.

This study proposes a novel type of bulk-cusp microstructure that enables multi-directional liquid self-spreading through precise geo-metric design without external energy input. As shown in Fig. 1a–c, cross- and square-shaped bulks are integrated with cusp elements through simple lithography, offering a scalable planar alternative to complex three-dimensional designs. The structure supports 5 distinct behaviors, including pinning, unidirectional, bidirectional, tri-directional and quad-directional spreading. Figure 1d illustrates that in cross-cusp designs, coupling and decoupling between the droplet body and the precursor film drive multi-directional spreading of both, whereas in square-cusp designs, the reduced coverage mainly guides the precursor film. Moreover, multiple spreading modes are achieved across a broad surface tension range of liquids. Localized capillary forces generated at narrow cusp gaps separate the precursor film and steer the main droplet body. These mechanisms are confirmed by computational fluid dynamics (CFD) simulations in combination with

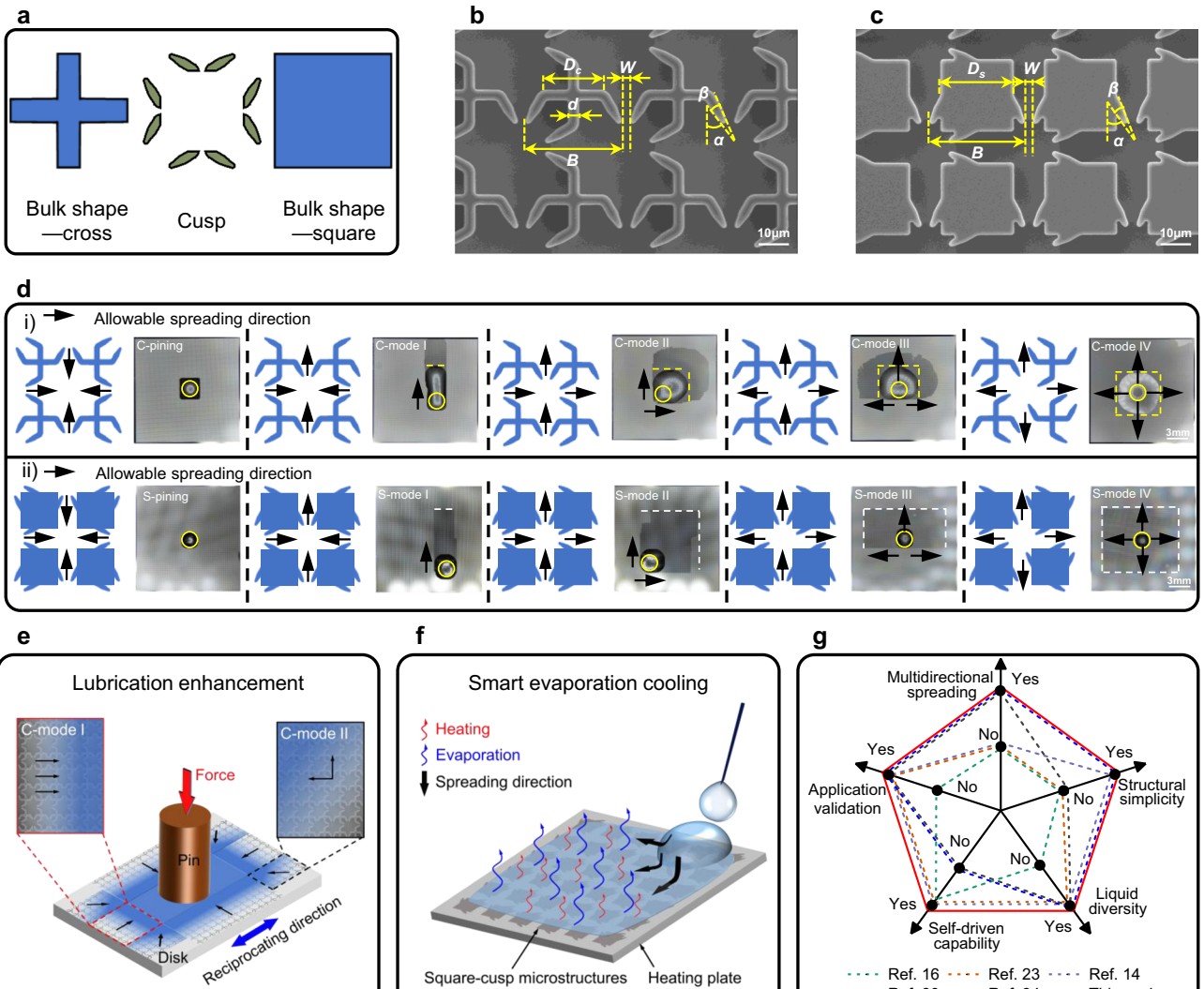

**Fig. 1 | Bulk-cusp microstructure for self-driven multi-directional liquid spreading. a** Design principle of the bulk-cusp microstructure, combining bulk geometries (cross and square) with cusps to regulate precursor film and droplet spreading. **b, c** SEM images of the cross-cusp and square-cusp microstructure with key geometric parameters. **d** Five representative spreading behaviors on cross-cusp (i) and square-cusp (ii) surfaces (yellow circle marks the droplet deposition position, yellow dashed lines indicate the advancing edge of the droplet body, and white dashed lines indicate the advancing edge of the precursor film). **e** Application

in mechanical lubrication enhancement, where the cross-cusp microstructure enables lubricant redistribution, improving surface lubrication and reducing fric-tion and wear. **f** Application in thermal management, where the square-cusp microstructure sustains precursor film supply, enhancing evaporation cooling efficiency. **g** Benchmarking comparison highlighting the advantages of the bulk-cusp design in multidirectional spreading, structural simplicity, liquid diversity, self-driven capability and application validation.

high-speed imaging. Beyond the spreading mechanism, the design exhibits dual adaptability, supporting rapid pump-free spreading as well as sustained operation under continuous injection. In lubrication, cross-cusp structures redistribute lubricant to the contact interface and reduce friction by about 35%, as demonstrated in Fig. 1e. For thermal management, Fig. 1f shows that square-cusp structures enable localized evaporative cooling with rapid and uniform temperature reduction. Finally, the benchmarking comparison in Fig. 1g highlights the unique combination of multidirectional spreading, structural simplicity, liquid diversity, self-driven capability and application validation, advancing the rational design of smart liquid-handling surfaces.

## Results

### Design of the bulk-cusp microstructure

The bulk-cusp microstructure includes two forms, cross-cusp and square-cusp, which influence the coupling or decoupling of the droplet body and precursor film (Fig. 1a). The detail design strategy and layout rules of the bulk-cusp microstructure are provided in Supplementary Note 1. An individual cross-cusp microstructure integrates a cross-shaped bulk with 4 directionally arranged cusps. The geometric parameters of a single cross-cusp microstructure are defined as cross length $(D_c)$ = 20 μm, cross width $(d)$ = 4 μm, cross-cusp length $(B)$ = 35 μm, cusp spacing $(W)$ = 3 μm, cusp-to-bulk angle $(\alpha)$ = 30°, and cusp apex angle $(\beta)$ = 15° (Fig. 1b). The spatial arrangement of these individuals dictates the droplet spreading behavior, forming 5 distinct array patterns, enabling droplet pinning (C-pinning), unidirectional spreading (C-mode I), bidirectional spreading (C-mode II), tri-directional spreading (C-mode III), and quad-directional spreading (C-mode IV), as shown in Fig. 1d and Supplementary Fig. S1. The microstructures were fabricated using photolithography, ensuring high precision and reproducibility. Scanning electron microscope (SEM) images in Supplementary Fig. S2 illustrate the structural variations of cross-cusp across different liquid spreading modes. In addition, a three-dimensional topographic reconstruction and cross-sectional profile of the quad-directional spreading microstructure reveal a structural depth $(h)$ of -12 μm, highlighting the uniformity of the fabricated features (Supplementary Fig. S3). Distinct from the cross-cusp configuration, the square-cusp microstructure (Fig. 1c) features a square-shaped bulk with a side length $(D_s)$ of 27 μm, primarily influencing precursor film dynamics. As illustrated in Supplementary Fig. S4, the directional arrangement of cusps enables the formation of 5 spreading modes, ranging from pinned precursor film (S-pinning) to controlled unidirectional, bidirectional, tri-directional, and quad-directional film spreading (S-mode I–IV). SEM images (Supplementary Fig. S5) further demonstrate that topology of the fabricated square-cusp microstructures. To further evaluate fabrication quality, Supplementary Fig. S6 shows narrow feature-size distributions centered at the design values, and Supplementary Fig. S7 indicates consistently low defect densities, together confirming that the photolithography process is precise, reproducible, and robust for large-scale fabrication.

### Multi-directional liquid spreading behavior

To evaluate the multi-directional liquid spreading performances of the bulk-cusp microstructures, the spreading behaviors of both the cross-cusp and square-cusp microstructure were systematically investigated in different modes. The relationship between the droplet morphology and the spreading characteristics was quantitatively analyzed through multi-directional liquid spreading experiments. As shown in Fig. 2a–e and Supplementary Movie 1, on the cross-cusp microstructure, the deionized (DI) water droplet remained pinned in the C-pinning mode without noticeable spreading, whereas in C-mode I–IV, the droplet body and precursor film spread unidirectionally, bidirectionally, tri-directionally, and quad-directionally, respectively. On the square-cusp

microstructure, the precursor film exhibited a spreading behavior similar to that of the droplet body on the cross-cusp microstructure, yet the droplet body itself remained nearly pinned (Supplementary Fig. S8, and Supplementary Movie 2). These results suggest that the local capillary force distribution, dictated by the cusp arrangement, plays a crucial role in guiding the multi-directional spreading of both the droplet body and precursor film.

Two complementary strategies were employed to quantify the spreading of the droplet body (Supplementary Note 2), namely directional length normalization and area-based equivalent radius analysis. All evaluations were conducted on the droplet body rather than the precursor film, as the latter was ultrathin and irregular, and its contour could not be reliably extracted in experiments. The spreading lengths $L_{x+}$, $L_{x-}$, $L_{y+}$, and $L_{y-}$ are defined as the maximum projected distances from the droplet center to the outer contour along the 4 Cartesian axes. At initial contact, a 5 μL droplet is approximated as an ideal sphere with radius $R$ (about 1 mm). To further evaluate irregular spreading areas accurately, 4 diagonal directions at 45°, 135°, 225°, and 315° were also considered, and the corresponding spreading lengths in the 4 directions were denoted as $L_1$, $L_2$, $L_3$, and $L_4$. All these values were normalized by the initial droplet radius $R$, giving the dimensionless ratio $L/R$. As shown in Supplementary Fig. S9, this eight-directional measurement provides a comprehensive evaluation of the spreading behaviors for both cross-cusp and square-cusp modes. Supplementary Fig. S10 and Fig. 2f reveal the time evolution and comparison of dimensionless spreading length on cross-cusp surfaces, where desired directions rapidly extend and stabilize at normalized lengths $(L/R)$ of about 2.6–4.5, while the non-guided ones remain near 1. In contrast, Supplementary Figs. S11 and S12a show square-cusp surfaces with nearly isotropic spreading, as all directions fluctuate slightly below 2. Velocity analysis in Fig. 2g and Supplementary Fig. S12b show that cross-cusp surfaces induce much faster dynamics, with instantaneous maximum speeds of about 13–26 mm/s aligned with desired directions, whereas square-cusp surfaces yield weaker responses with peak values generally below 6 mm/s. Normalized spreading lengths across eight directions are further compared in Fig. 2h, where cross-cusp surfaces exhibit pronounced elongation along allowable axes, whereas square-cusp surfaces maintain nearly isotropic patterns. These results demonstrate consistent directional alignment between spreading length and velocity, highlighting that the droplet body on cross-cusp surfaces exhibits much stronger anisotropy than on square-cusp surfaces.

To further characterize the overall multi-directional spreading behavior of the droplet body, the droplet spreading anisotropy factor $K$ is defined. For the Pinning mode and Mode IV, $K$ is defined as the average $L/R$ across all 4 directions (Eq. (1)):

$$K = \frac{L_{x+} + L_{x-} + L_{y+} + L_{y-}}{4R}, \text{Pining mode or Mode IV} \quad (1)$$

In Mode I–III, it is expressed as the ratio of the spreading lengths in forward directions to those in non-spreading directions. Consequently, the $K$ expression for Modes I–III is shown in Eq. (2):

$$K = \begin{cases} \frac{3L_{x+}}{L_{x-} + L_{y+} + L_{y-}}, \text{Mode I} \\ \frac{L_{x+} + L_{y+}}{L_{x-} + L_{y-}}, \text{Mode II} \\ \frac{L_{x+} + L_{x-} + L_{y+}}{3L_{y-}}, \text{Mode III} \end{cases} \quad (2)$$

As illustrated in Fig. 2i, the results reveal that C-mode I exhibits the highest $K$ value of 4.1, while in C-mode II, C-mode III, and C-mode IV, it gradually decreases to 3.1, 2.6, and 2.6, respectively, indicating that as the number of spreading directions increases, the liquid spreading is more uniform. In contrast, the square-cusp microstructure maintains $K$

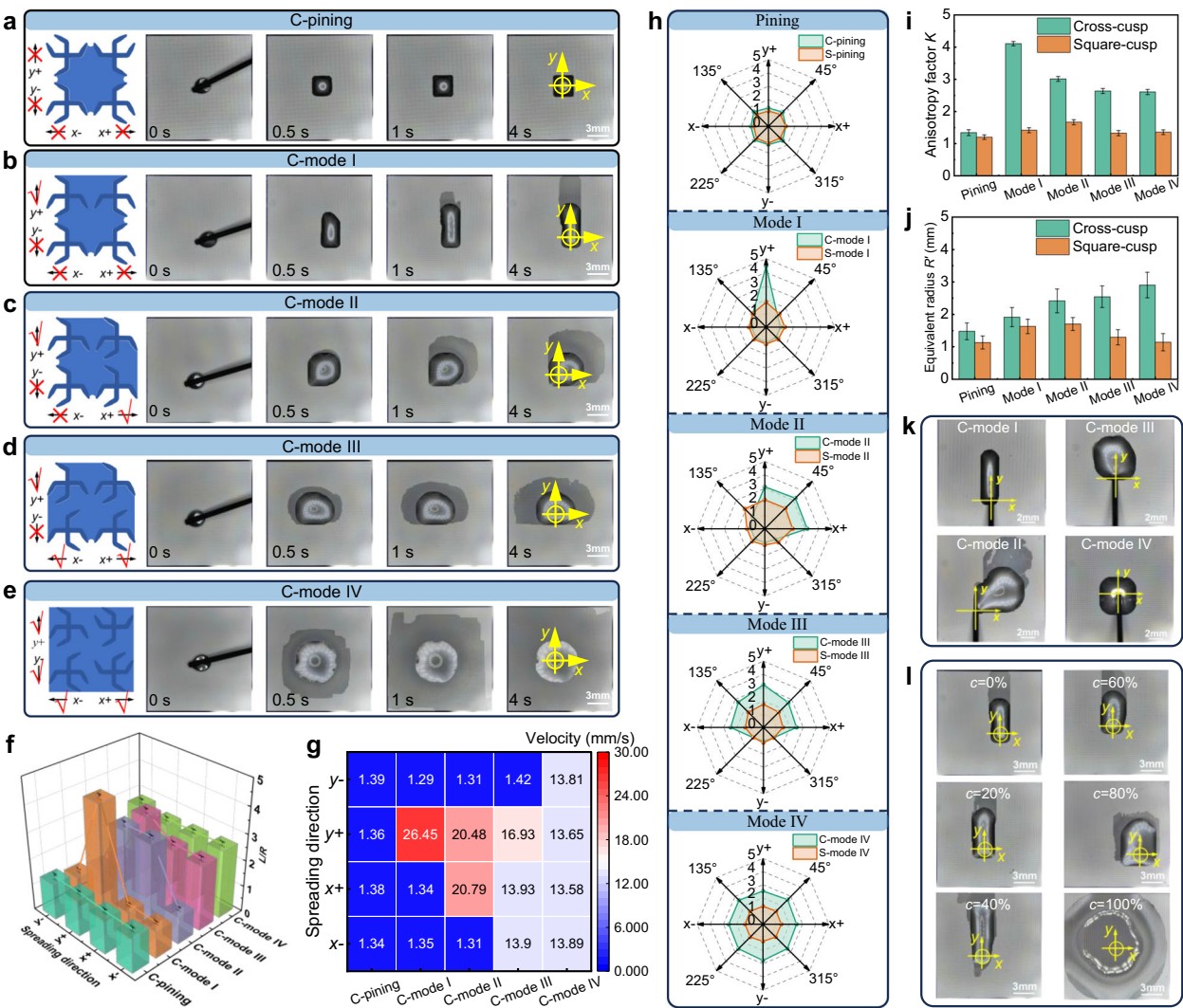

**Fig. 2 | Quantitative characterization of multi-directional liquid spreading behaviors on bulk-cusp microstructures.** Time-lapse images of droplet spreading on cross-cusp surfaces in C-pinning (**a**), C-mode I (**b**), C-mode II (**c**), C-mode III (**d**), and C-mode IV (**e**). (Yellow circles indicate the droplet deposition point.) **f** 3D bar chart of normalized spreading length $L/R$ along the 4 orthogonal directions ($x$-, $x$+, $y$- and $y$+) for different modes, showing pronounced anisotropy spreading on cross-cusp surfaces. Bars represent mean values, and error bars indicate mean ± SD ($n = 3$ independent experiments). **g** Heatmap of maximum instantaneous spreading velocity in 4 orthogonal directions for different modes, highlighting faster transport along guided axes on cross-cusp surfaces. **h** Radar plots comparing normalized spreading length in 8 directions for cross-cusp and square-cusp surfaces in different modes. **i** Statistical comparison of anisotropy factor $K$ between cross-cusp and square-cusp surfaces. Bars represent mean values, and error bars indicate mean ± SD ($n = 3$ independent experiments). **j** Equivalent spreading radius $R'$ in different modes, further quantifying the spreading areas. Bars represent mean values, and error bars indicate mean ± SD ($n = 3$ independent experiments). **k** Droplet morphologies in C-mode I–IV observed from top view for continuous injection, showing directional elongation along designed axes. **l** Spreading behaviors of droplets with different concentration $c$ ethanol–water mixtures (0–100% ethanol), illustrating the dependence of spreading behaviors on surface tension. Source data are provided as a Source Data file.

value close to 1 across all modes, further confirming its weak ability to influence multi-directional spreading of the droplet body.

To complement the direction-resolved length analysis, the global spreading was quantified by a quadrant-area method, and the plane was divided into 4 quadrants about the geometric center. The initial area in each quadrant is $S = \pi R^2/4$. After spreading, the areas of the 4 quadrants are $S_1$, $S_2$, $S_3$, $S_4$, and the normalized measures are $S_i/S$. An equivalent spreading radius is then defined as:

$$R' = \sqrt{\frac{S_1 + S_2 + S_3 + S_4}{\pi}} \qquad (3)$$

The scheme is shown in Supplementary Fig. S13. Time-dependent normalized spreading areas in 4 quadrants on different microstructures in 5 spreading modes are shown in Supplementary

Figs. S14 and S15. For cross-cusp surfaces, the $S_i/S$ reaches up to 12–13 in desired directions while remaining near 1–2 in non-desired directions, whereas on square-cusp surfaces it stabilizes around 1–6 in all quadrants. As summarized in Fig. 2j, the equivalent radius $R'$ on cross-cusps in spreading modes can exceed about $3R$, whereas that on square-cusps remains below about $2R$. These findings confirm that cross-cusp surfaces significantly amplify spreading along desired directions, while square-cusps preserve relatively weaker spreading behaviors.

To further assess whether such multi-directional spreading behavior can be maintained under continuous injection, additional experiments were carried out. DI water was continuously supplied using a microinjection pump at a rate of 10 μL/min, and the spreading dynamics were recorded. The results (Fig. 2k, Supplementary Figs. S16 and S17, and Movies 3 and 4) show that droplets extend

steadily along the designed pathways, with the spreading front remaining well guided throughout the process. These findings demonstrate that bulk-cusp structures enable reliable multi-directional spreading both in self-driven single-drop mode and under sustained pumping, providing stable adaptability for liquid handling.

To evaluate the universality of the design, ethanol–water mixtures with varying surface tension and dynamic viscosity were tested. As shown in Supplementary Fig. S18, the surface tension $\gamma$ decreases markedly with increasing ethanol concentration $c$, whereas the dynamic viscosity $\eta$ changes only slightly and the static contact angle $\theta$ decreases monotonically. Directional spreading was then examined on cross-cusp surfaces (Supplementary Fig. S19 and Fig. 2l). At ethanol concentrations up to 60%, droplets maintained stable multi-directional spreading along predefined directions, whereas excessively low $\gamma$ ($c \geq 80\%$) led to nearly isotropic wetting. Quantitative results (Supplementary Figs. S20 and S21) show that surface tension predominantly governs the stability and extent of directional spreading. To further assess liquids with high contact angle, ethanol–water mixtures were used on silicon wafers without normal plasma treatment. As summarized in Supplementary Fig. S22, low–contact-angle liquids enable efficient guided spreading, whereas high–contact-angle liquids progressively lose mobility until the droplets become pinned. Quantitative results (Supplementary Figs. S23 and S24) show that increasing contact angle reduces both normalized spreading length and velocity, thereby progressively suppressing anisotropic spreading. To elucidate the influence of viscosity, 4 liquids with distinct viscosities were tested in the C-mode IV configuration. As shown in Supplementary Fig. S25, the viscosities of these liquids span several orders of magnitude, while surface tension remains nearly constant. The spreading behaviors (Supplementary Fig. S26) reveal that low-viscosity liquids spread rapidly and uniformly, whereas high-viscosity liquids exhibit sluggish motion and limited coverage. Quantitative results (Supplementary Fig. S27) confirm that viscosity has little effect on spreading length but strongly reduces spreading speed, indicating that viscous resistance primarily governs the dynamics rather than the extent of multi-directional spreading. These results highlight that surface tension provides the driving force for multi-directional spreading, whereas viscosity primarily governs its dynamics, especially the transport speed.

## Mechanism of multi-directional liquid spreading

The multi-directional liquid spreading on the bulk-cusp microstructure involves the transport of both the droplet body and the precursor film, with the spreading direction being regulated by adjusting the arrangement of cusps. To elucidate the mechanism of directional liquid spreading on the bulk-cusp microstructure, a high-speed camera coupled with a microscope was employed to observe and analyze the spreading behavior. Taking the C-mode II surface as an example, as shown in Fig. 3a and Supplementary Movie 5, the narrow gaps formed between adjacent cusps generate capillary traction forces directed from the narrow to the wide end, facilitating the separation of the precursor film[36]. As the precursor film spreads and covers the microgrooves, it reduces the local contact angle, thereby inducing forward spreading of the droplet body. Due to the asymmetric distribution of capillary forces, both the precursor film and the droplet body undergo rapid spreading along the predetermined $x+$ and $y+$ directions. The precursor film spreads along the sidewalls and bottom surfaces of the bulk-cusp microstructure, where it is pulled by the surface tension $\gamma$ of the hydrophilic surfaces, resulting in a guided spreading behavior (Fig. 3b). Moreover, sequential snapshots in Supplementary Fig. S28 and Supplementary Fig. S29 further visualize the time-resolved evolution of precursor film in different spreading modes, confirming the critical role of cusp geometry in regulating directional spreading.

The directional spreading of the precursor film is determined by the force balance near the cusp region, as illustrated in Fig. 3c and Supplementary Fig. S30a. During forward spreading, the film is subjected to two capillary attraction components, namely $F_{d1}$ from the sidewalls and $F_{d2}$ from the bottom surface, as well as a backward resistance force $F_b$ induced by the sidewalls (Supplementary Note 3). The resultant force $F$ acting on the precursor film can be expressed as Eq. (4):

$$F = F_{d1} + F_{d2} - F_b = 2\gamma h \cos\theta \cos(\alpha - \beta) + \gamma l \cos\theta - 2\gamma h \sin(\alpha - \beta) \tag{4}$$

where $\gamma$ denotes the surface tension between the liquid and surface, $\theta$ is the intrinsic contact angle of the droplet, and $l$ is the local width of the precursor film at the cusp, respectively. For the precursor film to advance, the condition $F > 0$ must be satisfied, yielding the following design criterion:

$$\cos\theta > \frac{2h \sin(\alpha - \beta)}{l + 2h \cos(\alpha - \beta)} \tag{5}$$

This relation establishes that forward precursor film spreading is enabled only when the interplay between $\theta$ and $(\alpha - \beta)$ falls within a specific design range. In this study, the intrinsic contact angle $\theta$ is measured to be 35°. As shown by the red dot in the shaded region of Fig. 3d, this corresponds to the designed parameter point of the bulk-cusp microstructure. Notably, parameter combinations located closer to the coordinate origin yield stronger forward propulsion, thereby guiding the optimal geometrical design of bulk-cusp microstructure.

Due to the sharp-edge effect[37–39], the sharp edges of the cusps surrounding the bulk structure in the C-mode II surface can effectively pin the droplet body and restrict its spreading in the backward direction (Fig. 3f, g). In this case, the pinned liquid surface is assumed to be cylindrical, and the attractive forces between the liquid and the sidewalls as well as the bottom surface are taken into account. The analysis focuses on the capillary interactions resolved along the sidewall and bottom, as shown in Fig. 3e. For the case $\alpha \leq \theta < 90°$, the sidewall contribution is represented by an attractive term and the partial contact with the bottom adds an opposing horizontal term. The resultant capillary resistance against reverse motion is expressed as Eq. (6):

$$F' = F_b' - F_{d2}' = 2\gamma h \cos(\theta - \alpha) - \gamma l \tag{6}$$

To ensure effective backward pinning, the net resistance must satisfy $F' > 0$. For the case $\alpha \leq \theta < 90°$, this yields the geometric constraint, given by:

$$cos(\theta - \alpha) > \frac{l}{2h} \tag{7}$$

This inequation defines the condition for effective pinning of the precursor film. As shown in Supplementary Fig. S30b, the combinations of $\theta$ and $\alpha$ that satisfy the inequation ensure backward resistance. For $\theta = 35°$, the threshold is about $\alpha < 117.8°$, and the design parameters used in this study fall within the shaded region, confirming effective suppression of film retreat. The parameter points corresponding to the designed cusp geometry and the reference line of $\alpha = \theta$ are also marked, indicating that points closer to the $\alpha = \theta$ line experience stronger resultant pinning forces, thus enhancing the backward resistance effect.

To further validate the mechanism of cusp geometry guiding multi-directional spreading, CFD simulations were conducted using ANSYS Fluent 2022 R1 on the basic array units of both cross-cusp and square-cusp structures. For the cross-cusp arrays (Fig. 3h,

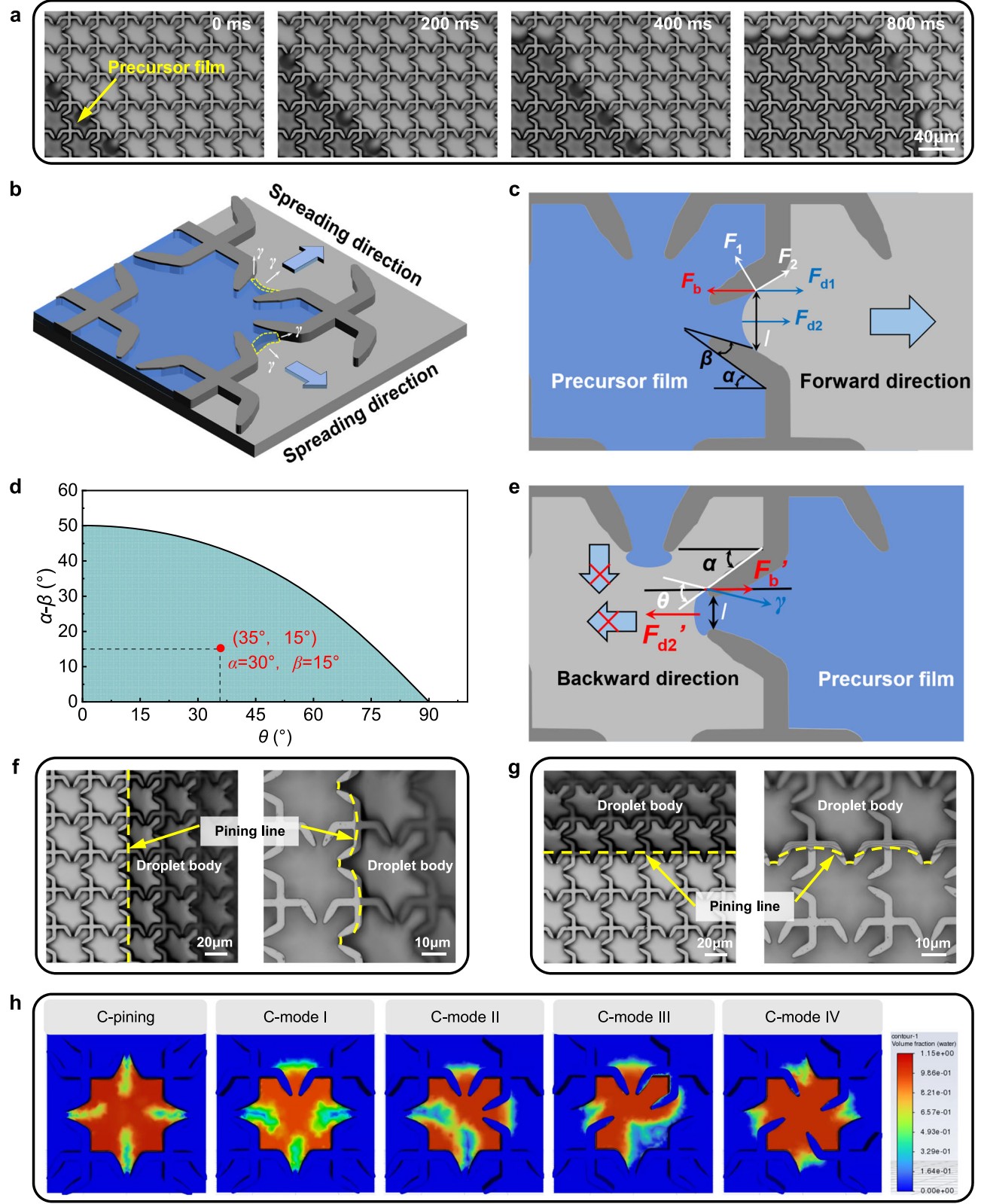

**Fig. 3 | Mechanism of multi-directional liquid spreading on the bulk-cusp microstructured surface. a** High-speed imaging sequences showing the spreading of the precursor film on a C-mode II surface. **b** Schematic illustrating precursor film spreading along the sidewalls and bottom grooves due to asymmetric capillary forces. **c** Force analysis near the cusp region, where forward traction forces $F_{d1}$ and $F_{d2}$ compete with backward resistance $F_b$. **d** Design parameter space defined by the intrinsic contact angle $\theta$ and cusp geometry $(\alpha - \beta)$, where the shaded region satisfies the condition for forward precursor film spreading; the red dot marks the

parameter set involved in this study ($\theta = 35°$, $\alpha = 30°$, $\beta = 15°$). **e** Schematic of backward force balance, where sidewall resistance $F_b'$ and the bottom component $F_{d2}'$ provide effective pinning that prevents film retreat. **f, g** Experimental observation and corresponding schematic of the pinning effect on the droplet body induced by sharp cusp edges geometry in the $x$ and $y$ directions. **h** CFD simulation of spreading morphologies in C-pinning and C-modes I–IV, demonstrating directional elongation of droplets along the designed axes, in close agreement with experimental observations. Source data are provided as a Source Data file.

Supplementary Fig. S31, and Movie 6) and the square-cusp arrays (Supplementary Fig. S32 and Movie 7), the simulations capture the droplets advancing preferentially along the desired pathways, consistent with the experimental observations. These results confirm that the designed geometries reliably dictate spreading pathways in each mode.

Overall, in bulk-cusp microstructure, the cooperative spreading behavior of the droplet body and precursor film is determined by the area coverage ratio of open regions defined by the geometric structure of the surface. Mechanism analysis shows that capillary forces generated between adjacent cusps trigger the separation and guided extension of the precursor film, while the coverage ratio of the area accessible to the precursor film determines the efficacy of the traction force in propelling the droplet body. Quantitative image analysis reveals that the cross-cusp microstructure possesses a relatively high open-area ratio (about 75%), compared to about 45% in the square-cusp counterpart. A higher open-area ratio facilitates precursor film spreading by providing continuous capillary pathways and expands wettable interfaces, thereby enabling effective traction to drive the droplet body along desired directions. Consequently, the droplet body on cross-cusp surfaces exhibits rapid, anisotropic spreading with high directional fidelity. In contrast, the smaller open-area ratio of square-cusp surfaces limits precursor film connectivity, leading to fragmented film coverage, weaker traction on the droplet body, and restricting directional control primarily to the precursor film. These results demonstrate that adjusting the coverage ratio of precursor film offers a viable strategy to decouple and independently control the directional spreading of the droplet body and precursor film, thereby inspiring the design of smart surfaces with controllable multi-directional liquid spreading.

## Application in lubrication enhancement

Stable lubrication is essential for reducing friction and wear in mechanical systems, yet conventional surface texturing strategies often rely on surface grooves or dimples within the contact region, which may lack long-term supply efficiency due to the continuous wear. The cross-cusp microstructures enable controllable droplet spreading and facilitate multi-directional liquid transport without external energy input. Leveraging this characteristic, the microstructure was applied to the non-contact region surrounding the tribo-pair to guide liquid spreading along the periphery. This design promotes sustained lubricant supply to the friction contact region and improves overall lubricant utilization. As shown in Fig. 4a, C-mode I and C-mode II microstructures are alternately arranged around the frictional interface, forming a guided pathway that continuously channels lubricant toward the contact zone. The combination of C-mode I and C-mode II spreading modes enables effective delivery and redistribution of lubricant without interfering with the load-bearing surface. To further clarify the specific contribution of cusp features, a surface with cross microstructures but without cusps was fabricated and tested alongside the bare and cross-cusp surfaces, with the characterizations of these three structures summarized in Supplementary Fig. S33.

Tribological performance was assessed under water-lubricated conditions using a pin-on-disk tribometer (Supplementary Fig. S34a). As shown in Fig. 4c, under a load of 30 N and sliding frequency of 4 Hz, the friction coefficient of the bare surface increases from about 0.16 to 0.30 with strong fluctuations, while the cross surface without cusps shows a similar trend, stabilizing around 0.28–0.30. In contrast, the cross-cusp microstructured surface maintains a much lower and steadier coefficient of about 0.20 throughout the test. Across a wide range of normal loads (10-50 N, 4 Hz, Supplementary Fig. S34b) and sliding frequencies (2-6 Hz, 30 N, Supplementary Fig. S34c), the cross-cusp surface consistently shows the lowest friction, with reductions of about 15–35% compared to the bare case. As demonstrated in Fig. 4c,

post-test surface morphologies reveal severe grooving and material tearing on the bare specimen, and the cross surface without cusps also shows scratches and partial wear, whereas the cross-cusp surface largely retains a smooth and intact morphology with only shallow scars. This lubrication improvement is attributed to the continuous spreading of lubricant from the periphery toward the contact region, driven by the capillary-guided spreading enabled by the cross-cusp microstructure. The established liquid pathways ensure continuous wetting at the contact interface, stabilizing lubrication in dynamic conditions. In addition, recent progress in advanced lubrication strategies, including protective coatings[40,41], lubricant additives[42–45], direct surface texturing[46–49], and surface strengthening[50–53], is summarized in Supplementary Note 4 and Fig. S35. Reported friction reduction rates of these approaches generally remain below 30%. However, the present design achieves about 35% reduction, surpassing existing strategies by enabling passive, capillary-driven lubricant replenishment from the non-contact region. This structurally simple and scalable approach therefore represents an innovative and energy-efficient route for advanced lubrication enhancement.

## Application in thermal management

In high-flux thermal systems and scenarios demanding localized heat dissipation, achieving efficient and long-lasting cooling remains essential for operational reliability. The square-cusp microstructure enables directionally guided precursor film spreading without external energy input, establishing rapid and continuous wetting pathways that lay the foundation for enhanced evaporation-based cooling. As shown in Fig. 4d and Supplementary Movie 8, upon deposition of a 5 μL droplet of deionized water, the precursor film initially spreads along the positive x-direction in the S-mode I region and subsequently redirects along the negative y-direction in the S-mode II region. This guided spreading enables a reorientation of the spreading pathway, leading to full-area coverage within ~20 s.

To evaluate the practical cooling performance, evaporation-based cooling experiments were conducted by dripping identical volumes of deionized water on the S-mode II microstructured surface, a square microstructured surface without cusps, and the bare silicon surface, with their morphologies shown in Supplementary Fig. S36. Infrared thermography was used to monitor real-time temperature evolution. As shown in Fig. 4e and Supplementary Movie 9, following a single 5 μL droplet injection, the bare surface exhibits only localized temperature reduction beneath the droplet, with the average temperature decreasing from about 49 °C to 42 °C within 100 s. The square microstructured surface provides limited improvement, reducing the average temperature to about 36 °C over the same period, with the cooling distribution remaining non-uniform. In contrast, the square-cusp surface shows a rapid and uniform temperature drop across the entire structure, with the average surface temperature decreasing from about 49 °C to 27 °C within only 60 s. These differences are further quantified by the temperature evolution curves in Fig. 4f, which clearly highlight the superior temperature reduction efficiency of the square-cusp design compared with the other two surfaces. Furthermore, in a repeated injection scenario where 5 μL droplet was dripped every 30 s, the three surfaces display distinct thermal responses. On the bare surface, temperature fluctuates significantly between injections, and localized hot spots frequently emerge, consistent with Supplementary Fig. S37 and Supplementary Movie 10. The square microstructured surface partly improves uniformity but still exhibits clear fluctuations and fails to maintain a narrow stable range. The square-cusp microstructured surface achieves the most stable performance, where the cooling zone is enlarged and the temperature remains confined within a narrow band throughout the cycles. Curves in Fig. 4g confirmed these observations. The bare surface maintains the highest temperature of about 38–44 °C with the highest variation. The square microstructured surface lies in an intermediate regime of

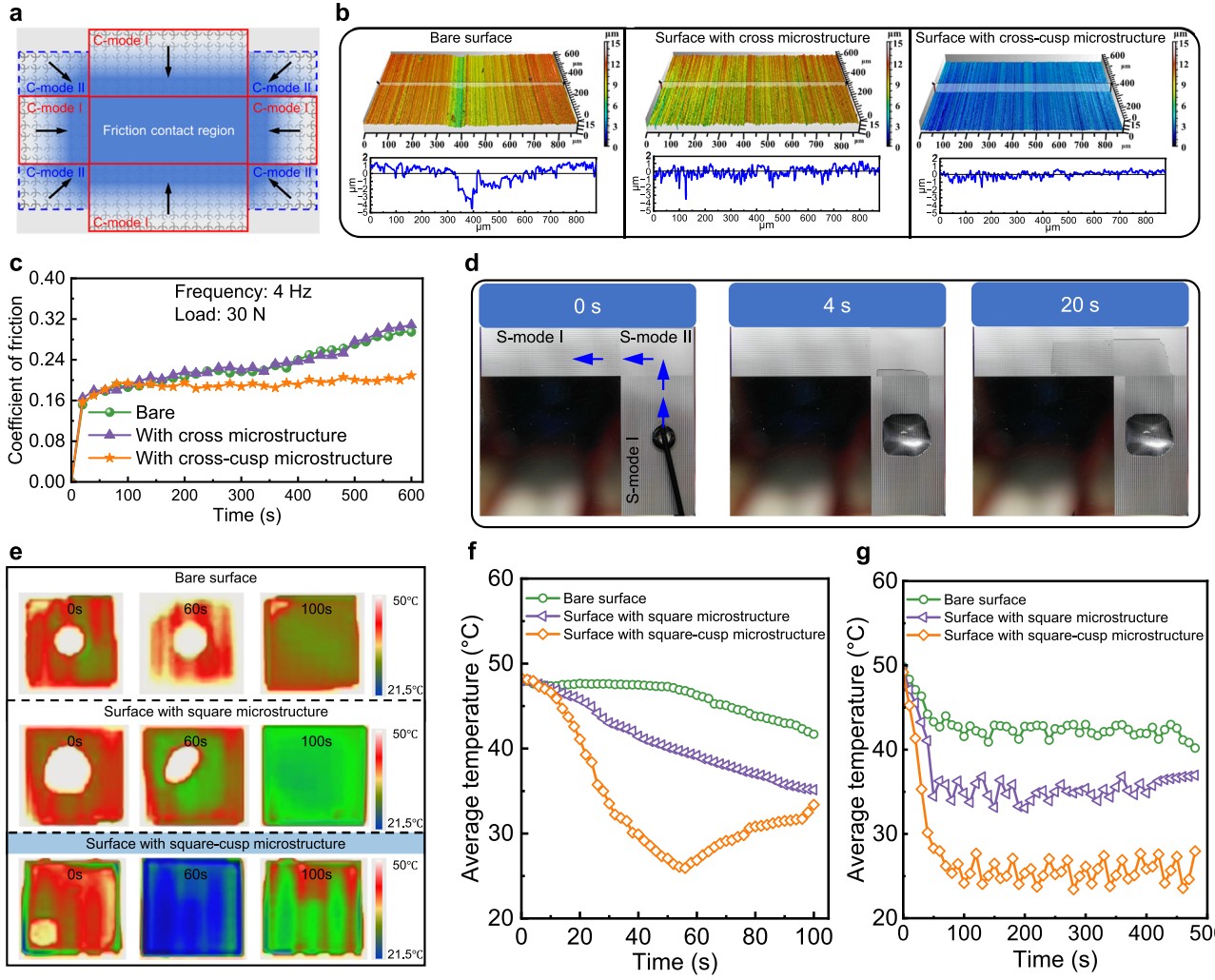

**Fig. 4 | Lubrication enhancement and thermal management enabled by the bulk-cusp microstructure. a** Schematic of cross-cusp microstructures (C-mode I and II) arranged around the non-contact zones of a friction interface to guide lubricant redistribution. **b** Post-test wear morphologies of the bare surface, the cross microstructured surface, and the cross-cusp microstructured surface, showing severe abrasion on the bare surface, partial mitigation on the cross microstructure, and negligible wear on the cross-cusp surface. **c** Friction coefficient evolution under 30 N and 4 Hz, where the cross-cusp microstructured surface maintains the lower and more stable values compared with the bare and cross microstructured surfaces. **d** Directional spreading of a precursor film from S-mode I to S-mode II regions, forming a continuous wetting pathway. **e** Infrared thermal images after deposition of a single 5 μL droplet, comparing the bare surface, the square microstructured surface, and the square-cusp microstructured surface, with the latter showing the fastest and most uniform cooling. **f** Average temperature decrease over time corresponding to (**e**), highlighting effective cooling on the square-cusp surface. **g** Temperature curves under repeated droplet addition every 30 s, showing unstable cooling on the bare surface, moderate improvement on the square microstructured surface, and consistently low and stable values of 24–26 °C on the square-cusp surface. Source data are provided as a Source Data file.

roughly 33–37 °C with moderate fluctuations. The square-cusp microstructured surface stabilizes at the lowest level of 24–26 °C with minimal variation. These results collectively demonstrate that the square-cusp microstructures sustains both low and stable temperatures under repeated liquid supply.

The above results demonstrate that the square-cusp microstructures facilitate effective, long-lasting and uniform evaporation cooling by leveraging precursor film guidance to expand the effective wetting area and promote distributed evaporation. As summarized in Supplementary Note 4, unlike conventional strategies based on surface modifications[54,55], nanocoatings[56–58], capillary wicks[59–61], or external energy inputs[62–64], the square-cusp design enables efficient heat dissipation, offering a promising non-invasive solution for localized thermal management in advanced heat dissipation systems.

## Discussion

A structurally simple and functionally controllable bulk-cusp microstructure has been demonstrated to enable self-driven, multi-directional liquid spreading with independently regulated transport of the droplet body and precursor film. By adjusting the cusps orientation and bulk shapes, liquid spreading in 0 to 4 directions can be controlled without external energy input and is valid under both single-drop deposition and continuous injection. The cross-cusp microstructure, featuring a high coverage ratio of the open area, generates sufficient capillary traction to drive the precursor film and droplet body. While the square-cusp microstructure, with a lower coverage ratio, primarily guides the precursor film. Mechanism analysis reveals that the capillary forces generated between adjacent cusps initiate the separation and directional spreading of the precursor film, while the coverage ratio of the precursor film determined whether the droplet body can be dragged forward. The controllable spreading behaviors of the bulk-cusp microstructure have been validated in two functional scenarios. In lubrication, the cross-cusp microstructures placed in non-contact regions enabled directional lubricant redistribution, effectively reducing friction and wear, with a friction reduction of about 35% compared with the bare surface. In thermal

management, square-cusp surface establishes continuous precursor film pathways for efficient evaporative cooling, achieving rapid temperature reduction and maintaining a stable low temperature under repeated liquid supply. Overall, these findings expand the design strategy for controllable multi-directional liquid spreading, offering a new avenue for designing smart surfaces with liquid-handling surfaces.

## Methods

### Materials
Ethanol–deionized water mixtures with various volume concentrations (0%, 20%, 40%, 60%, 80%, and 100%) were prepared for the experiments. Absolute ethanol (analytical grade) was purchased from Shanghai Macklin Biochemical Co., Ltd (Shanghai, China), and deionized water was supplied by Shanghai Aladdin Biochemical Technology Co., Ltd (Shanghai, China). In addition, three oils with different viscosities were selected for comparison, including white oil, paraffin oil, and castor oil, all of which were obtained from Shanghai Aladdin Biochemical Technology Co., Ltd. All reagents were of analytical grade and were used without further purification.

### Fabrication of bulk-cusp micro-structured surfaces
The bulk-cusp micro-structured surfaces were fabricated on silicon wafers using standard photolithography (Guangzhou Zetian Electronics Co., Ltd, China). Prior to lithography, the silicon wafers were first immersed in a freshly prepared piranha solution ($H_2SO_4:H_2O_2 = 7:3$) for 30 min to remove organic contaminants, followed by sequential rinsing with deionized water, ethanol, and acetone, and subsequently baked at 100 °C for 10 min to eliminate residual moisture. A positive photoresist (AZ4620, MicroChemicals, Germany) was then spin-coated at 3000 rpm using a Laurell WS-650-23 coater, and prebaked in two steps (5 min at 60 °C and 20 min at 90 °C) to improve adhesion and coating uniformity. UV exposure was carried out using a SUSS MA6 mask aligner, and the patterned resist was developed in AZ-400K solution. Deep reactive ion etching was performed to obtain the bulk-cusp microstructure, after which the residual resist was completely removed by rinsing with acetone, ethanol, and deionized water, yielding sharply defined cusp features.

To achieve suitable wettability for directional spreading, the as-fabricated surfaces were treated in a plasma cleaner (Pluto-T, PLU-TOVAC, Shanghai, China) at a power of 100 W for 10 s, converting the initially hydrophobic surfaces into hydrophilic ones. Before plasma treatment, the contact angle of the micro-structured surface was ~95°, which decreased to around 15° immediately after treatment. The treated surfaces were then exposed to ambient air (25 °C, 30% relative humidity) for 48 h, during which the contact angle gradually increased to ~35°. This intermediate wettability condition was favorable for enabling multi-directional spreading of water droplets on the bulk-cusp microstructures. Contact angle measurements were performed using a contact angle goniometer (OCA20, Dataphysics Co., Germany).

### Multi-directional liquid spreading experiments
A micropipette was used to extract a defined volume of deionized water, and ~5 μL of the liquid was dispensed onto the micro-structured surface as a single droplet. In addition to deionized water, ethanol–water mixtures with volume concentrations of 20%, 40%, 60%, 80%, and 100% were tested to examine the influence of surface tension reduction, and lubricating oils with different viscosities (white oil, paraffin oil, and castor oil) were employed to evaluate viscosity-dependent spreading dynamics. All liquids were dispensed in the same volume of 5 μL using calibrated micropipettes to ensure consistency. An optical camera (IMX334, Sony Corporation, Japan) captured images and videos of the spreading dynamics of both the droplet body and the

precursor film on the surface. To visualize the detailed multi-directional spreading behavior, a high-speed camera (AZ9501, AZ Instruments Corporation, Taiwan, China) was mounted on a microscope (RX50M, China) to record the rapid spreading processes of the droplet body and precursor film. Moreover, to better simulate practical liquid transport scenarios, a continuous injection condition was introduced, where a microinjection pump (ZS100-01, Baoding Chuangrui Precision Pump Co., Ltd, China) delivered deionized water at a constant rate of 10 μL/min. This enabled direct comparison between instantaneous spreading after single-drop deposition and steady-state dynamics under sustained inflow. To ensure reliability and reproducibility, each experimental condition was repeated three times. The recorded spreading velocities and normalized spreading lengths were averaged, and error bars represented standard deviation (SD). All experiments were conducted under still-air laboratory conditions.

### Tribological experiments
A 45# steel disk (surface roughness 0.5 μm, hardness HRC48) was selected as the lower friction pair. Four grooves with the same depth as the cross-cusp micro-structured silicon wafer was milled around the central contact region. A copper pin (surface roughness 0.5 μm, hardness HRC30) with a diameter of 6 mm served as the upper friction pair. Before testing, the contact surfaces of both friction pairs were polished and cleaned to remove residual contaminants, after which the silicon wafer samples were embedded into the grooves and fixed with Polydimethylsiloxane (PDMS) to ensure surface flushness. In addition to the bare sample without microstructures around the contact region, a cross-patterned surface without cusps was also tested as a control to further clarify the role of cusp features. An equal volume (50 μL) of deionized water was applied as a lubricant at the same location on both the bare surface and the microstructured surface with non-contact zone placement. Tribological experiments were conducted using a reciprocating tribometer (WDT-1, Shandong Huaqi Instrument Co., Ltd, China), and the real-time coefficient of friction were recorded via the accompanying software. The operating conditions included three sets of scenarios: (i) a test at 30 N load and 4 Hz frequency for 10 min; (ii) load variation at a fixed frequency of 4 Hz with normal loads of 10, 20, 30, 40, and 50 N; and (iii) frequency variation at a fixed load of 30 N with sliding frequencies of 2, 3, 4, 5, and 6 Hz. Test for each condition was lasted for 10 min and was repeated three times under identical conditions, and the reported coefficient of friction represented the average of the measurements with error bars corresponding to standard deviation (SD). All experiments were conducted under still-air laboratory conditions.

### Thermal management experiments
The prepared square-cusp micro-structured silicon wafer was placed on a temperature-controlled heating plate maintained at 50 °C. The heating platform was a microcomputer-controlled unit (JF-956B, JFTOOIS, China), ensuring accurate and stable thermal conditions throughout the tests. A flat silicon wafer without microstructures served as the comparison, with all other treatments kept identical. In addition, a square-patterned surface without cusp was also included as a control, allowing further evaluation of the specific contribution of cusp features to thermal performance. Prior to each test, all silicon wafer samples were thoroughly cleaned with ethanol and deionized water, followed by nitrogen drying, to eliminate any residual dust or contaminants. For single-drop evaporation experiments, ~5 μL of deionized water was dispensed onto the sample surface using a micropipette. For continuous evaporation experiments, the same volume of water was added to the surface about every 30 s. During the droplet addition and evaporation processes, an infrared thermal camera (HM-TPK20-3AQF/W, HIKMICRO, China) continuously recorded the temperature evolution of both the

microstructured and bare surfaces. The infrared images were further processed using custom code to extract the spatially averaged surface temperature in real time. To ensure accuracy and reproducibility, each experimental condition was repeated three times with independently prepared samples. All experiments were conducted under still-air laboratory conditions.

**Contact angle, surface tension, and viscosity measurements**

Contact angle measurements were performed using a contact angle goniometer (OCA20, Dataphysics Co., Germany). Surface tension of the liquids was measured with a tensiometer (DSA100S, KRUSS, Germany), while dynamic viscosity was determined using a rotational viscometer (DV2T, Brookfield, USA). For each liquid or surface type, three independent measurements were carried out. Specifically, contact angle measurements were performed on three droplets placed at separate positions on the wafer, and each droplet was measured three times to minimize random error. Similar procedures were followed for both surface tension and viscosity tests to ensure consistency. The measurement results were presented as the mean values, with error bars representing the standard deviation (SD). All measurements were conducted under still-air laboratory conditions.

**Morphology characterization**

The surface morphology of the bulk-cusp microstructures was characterized using a tungsten filament scanning electron microscope (EVO10, ZEISS, Germany) after sputter-coating with a thin layer of gold. For SEM imaging, multiple regions across each wafer were examined to verify the uniformity of microstructure morphology, and representative images were selected from consistent observations. The three-dimensional surface morphology of the microstructures and the wear profiles of the samples after tribological tests were characterized using a laser confocal microscope (DCM8, Leica, Germany). All results were summarized as mean values with standard deviations (SD) to account for sample-to-sample variations.

## Data availability

All relevant data supporting the key findings of this study are available within the paper and its Supplementary Information or from the corresponding author upon request. Source data are provided with this paper.

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

## Acknowledgements

The authors are thankful for the financial support from the National Natural Science Foundation of China (52575228 and 52075418). The authors would like to thank the instrument analysis center of Xi'an Jiaotong University for their assistance with morphological characterization.

## Author contributions

S.J. Dai led the study, including methodology, major experiments, data analysis, and preparation of the original manuscript draft. H.Z. supervised the entire project, proposed original conception, guided methodology and interpretation, reviewed and refined the manuscript, and provided funding support. Y.L. assisted with experimental arrangement, data acquisition, and analysis. F.H.Y. contributed to simulation, modeling, and assisted with figure preparation. K.B.S. supported literature survey and supplementary experimental planning. G.N.D. reviewed the manuscript, and offered constructive comments.

## Competing interests

The authors declare no competing interests.
