## [Transparent Peer Review file · Nature Communications]

Bulk-cusp microstructure for controllable multi-directional liquid spreading

Corresponding Author: Professor Hui Zhang

Version 0:

Reviewer comments:

Reviewer #1

(Remarks to the Author)

The concept of directional liquid spreading based on cusp structures is interesting. However, I find that the current manuscript does not meet the standards expected for Nature Communications, and I do not recommend it for publication in its present form.

Major Comments:

1. The authors only tested a single liquid (water) with one type of microstructure. To validate the proposed design criteria (Eq. 5 and Eq. 6), a broader set of experiments is needed. Specifically, how do key liquid properties—such as surface tension and viscosity—affect the directional spreading? Additional combinations would be necessary to support the universality of the proposed model.
2. Spreading velocity is a critical parameter for many practical applications. The manuscript lacks quantitative results on spreading dynamics, which are essential for evaluating performance. The authors should report and discuss the spreading velocity.
3. The authors propose applications in lubrication and thermal management. However, comparing their surface with a bare (non-structured) surface is insufficient. To convincingly demonstrate the contribution of directional spreading, a proper control—such as a surface with similar microstructures but without cusp features—should be used.
4. In Equation 6, the contribution of the bottom surface is not included. A justification or correction is needed to ensure completeness and consistency in the physical modeling.
5. It is unclear whether directional spreading occurs under continuous water supply or only for a single droplet. The latter would significantly limit practical applications. The authors should clarify this and ideally provide experimental evidence under continuous conditions.

Minor Comments:

1. The term “fluid spreading” should be revised to “liquid spreading” throughout the manuscript, as the study focuses specifically on liquids.
2. On Page 2, line 40: The statement “liquid spreading from high to low surface energy regions” is incorrect. In fact, liquids spread from regions of low to high surface energy. This should be corrected

Reviewer #2

(Remarks to the Author)

In this article, a novel type of microstructured surface, fabricated using standard photolithography, is proposed to achieve controllable, multidirectional fluid wetting.

The authors experimentally demonstrate that the proposed three-dimensional microstructures manipulate capillary forces to guide liquid spreading along specific directions without any external energy input.

I find the topic exciting, and both the research work and methodology are robust. This contribution advances our understanding of next-generation smart surfaces for a wide range of applications.

I recommend the publication of this article in Nature Communications, with only a few minor comments the authors may wish

to consider:

1. Do the square-cusp microstructures have the same depth as the cross-cusp ones? This is not mentioned in the article.
2. Labels should be added to all panels of Figures S8 and S9 to indicate the corresponding mode.
3. In Equation 4, the term $\cos\theta$ appears on both sides of the equation. It would be clearer to solve the equation with respect to $\cos\theta$, so it appears only on the right-hand side.
4. Since it is suggested that these microstructures could be used in advanced tribological systems, it would be useful to comment on their durability. In such applications, how would these surfaces behave in the presence of debris accumulation? Complex surface features might trap particles and accelerate wear.
5. Particles may also originate from dusty or dirty environments. How would the presence of such particles affect the efficiency of the proposed microstructures in general?
6. The article mentions that the intrinsic contact angle of the surface is 35° . How sensitive is the behavior of the proposed microstructures to surface wettability? For example, does their efficiency change significantly when using liquids with different surface tensions?

Reviewer #3

(Remarks to the Author)

This paper presents an interesting investigation of bulk-cusp microstructures for controlling multi-directional fluid spreading, with potential applications in lubrication and thermal management. While the structures are very novel and the topic is of interest, the manuscript requires substantial improvements before it would be suitable for publication in a high-impact journal such as Nature Communications.

Major Concerns:

1. The manuscript claims a novel design, but the introduction cites numerous existing approaches to directional fluid control. The authors should consider making a stronger case for how the proposed bulk-cusp design represents a significant advance over existing microstructures. What specific limitations of prior work are overcome, and what new capabilities are demonstrated? A more thorough comparison to the existing literature is required, with quantitative metrics to support claims of improved performance.
2. The proposed mechanism of fluid spreading relies on simplified force balance equations. A more rigorous model, potentially incorporating computational fluid dynamics (CFD) simulations, would be beneficial to validate the proposed mechanism and provide deeper insights into the underlying physics.
3. The experimental section needs more detail regarding the setup, error analysis, and statistical significance of the results. The authors should provide more comprehensive characterization data for the fabricated microstructures, including information on feature size uniformity and defect density. Furthermore, the performance gains in lubrication and thermal management must be benchmarked against state-of-the-art technologies to quantify the improvement achieved by the proposed design.
4. The manuscript is generally well-written, but some sections could be clarified. The distinction between cross-cusp and square-cusp structures, as well as their roles, should be emphasized.
5. Figures should be improved with labels and self-explanatory captions. It would be beneficial if authors present the exact figure in the main text, along with data and schematic illustrations from the same structures used (with the same orientation and spreading mode). For example, Figures 3a, d, and e have the same orientation of cups, but Figure 3b has a different one.
6. The method for quantifying the spreading length ('L') requires significant clarification. As defined, 'L' represents the spreading length along the x_+ , x_- , y_+ , or y_- axes. However, this definition is unclear in cases where spreading is not strictly unidirectional, such as the quasi-circular spreading observed with cross-cusp structures and the square-like spreading with square-cusp structures. The manuscript lacks a precise explanation, with illustrative examples, of how 'L' is determined for each spreading mode (C-pinning, C-mode I-IV, S-pinning, S-mode I-IV).

The authors are advised to:

- a. Provide an unambiguous definition of how 'L' is measured for each spreading mode, including a diagram illustrating the measurement process for each distinct case. Additionally, which droplet radius is considered? Of a spread droplet or one calculated based on the droplet volume.
- b. Consider alternative measurement approaches that are less sensitive to the shape of the spreading pattern. For instance, an effective radius (based on the spread area) or a direct area measurement might be more appropriate for non-unidirectional spreading (e.g., $A_{\text{drop}}/A_{\text{spread}}$ where A_{drop} is the base area of droplet and A_{spread} is the spreading area).
- c. Provide an unambiguous definition of how 'L' is measured for each spreading mode, including a diagram illustrating the measurement process for each distinct case.
- d. Consider alternative measurement approaches that are less sensitive to the shape of the spreading pattern. For instance, an effective radius (based on the spread area) or a direct area measurement might be more appropriate for non-unidirectional spreading.
- e. Justify the chosen measurement metric, explaining why it is the most relevant for characterizing the fluid spreading behaviour under investigation, particularly concerning the goal of directional control. If square spreading is observed, its

functional significance should be articulated.

Minor comments and suggestions:

1. In Figure 1a, the legend for the illustration will read more easily if it goes in order: cross, cusp, and square.
2. In Figure 3a, when the liquid is spreading, there is a change in the meniscus that resembles the bubble or meniscus deformation in the spreading direction. Can authors comment on it and how it looks for different structures?

Version 1:

Reviewer comments:

Reviewer #1

(Remarks to the Author)

The authors have revised the manuscript substantially in response to my previous comments. Considering the scope of the additional experimental results and the validity of the claims supported by them, I believe the manuscript has been significantly improved and can now be considered for publication in Nature Communications.

Reviewer #2

(Remarks to the Author)

The science in this manuscript is strong, and the authors have clearly made considerable effort to address the concerns raised in the first round of review. However, I note that the current version of the manuscript has become quite long and dense. The main text contains four figures with many subpanels, and the supplementary material now spans over 50 pages with a large number of additional figures. Many of these supplementary figures are referenced in the main text, which makes it challenging to follow the narrative smoothly.

While the additional data may be scientifically valid, the sheer volume of figures and supplementary material makes the manuscript harder to read and digest. I would encourage the authors to consider ways to streamline the presentation—perhaps by consolidating subpanels, combining figures where appropriate, or ensuring that only essential supplementary figures are cited in the main text. This would improve clarity and help readers focus on the key findings without compromising the scientific rigor of the work.

Overall, the manuscript is scientifically sound, but a more concise presentation would make it more accessible and impactful.

Reviewer #3

(Remarks to the Author)

I appreciate the authors' careful and thorough revision of the manuscript. They have addressed all concerns previously raised with detailed explanations, additional analysis, and clearer presentation of results. The overall quality of the paper has improved substantially, both in terms of clarity and scientific rigor. The discussion is now more comprehensive and well-balanced, effectively situating the findings within the existing literature. I believe the manuscript is now in a publishable form and recommend it for acceptance.

Response to Comments

Dear Editor and Reviewers:

Firstly, thanks for the comments and queries! We deeply appreciate the earnest comments given by you all. They are very valuable and helpful for our improvement of the quality of this manuscript. We thus, to our best ability, carefully addressed each point and revised the manuscript accordingly. The revised portion is marked in red in the revised manuscript and Supporting Information. Below are our responses to the reviewers' comments:

Reviewer #1:

The concept of directional liquid spreading based on cusp structures is interesting. However, I find that the current manuscript does not meet the standards expected for Nature Communications, and I do not recommend it for publication in its present form.

Major Comments:

1. The authors only tested a single liquid (water) with one type of microstructure. To validate the proposed design criteria (Eq. 5 and Eq. 6), a broader set of experiments is needed. Specifically, how do key liquid properties—such as surface tension and viscosity—affect the directional spreading? Additional combinations would be necessary to support the universality of the proposed model.

Response 1: We sincerely thank the respected reviewer for this insightful comment. The suggestion to investigate the effects of liquid properties such as surface tension and viscosity on directional spreading is highly valuable. We agree that expanding the experimental scope beyond a single liquid and a single type of microstructure is important to validate the universality of the proposed design criteria and to strengthen the robustness of our model.

To study how liquid properties affect directional spreading, we firstly prepared ethanol–deionized (DI) water mixtures at six volume fractions—0% (pure DI water), 20%, 40%, 60%, 80%, and 100% (pure ethanol)—and measured their spreading behaviors on identically plasma-treated substrates. This is a typical method to study the

influence to liquid surface tension by many other researchers, especially in the field of liquid wetting behaviors. All microstructured samples were processed in a plasma cleaner (Pluto-T, PLUTOVAC, Shanghai, China) at 100 W for 10 s to convert the initially hydrophobic surface (static contact angle (CA) $\approx 95^\circ$ before treatment) into a hydrophilic one. The treated samples were then exposed to ambient air for 48 h before testing. **Figure R1** summarizes the liquid properties and wetting. The surface tension γ decreases markedly with increasing ethanol content (72 to 22 mN m⁻¹), while the dynamic viscosity η varies only slightly (1 to 3 mPa • s). The measured static contact angle (CA) on the treated surface decreases monotonically with ethanol concentration (from $\sim 35^\circ$ for DI water to $\sim 0^\circ$ for ethanol). Given that η changes much less than γ across the tested range, the following analysis focuses on the effect of surface tension.

Figure R1. Variation of liquid properties (surface tension, dynamic viscosity, and static contact angle) changing with ethanol concentration. a Surface tension and dynamic viscosity of ethanol–water mixtures varying with ethanol concentration. **b** Static contact angle of ethanol–water mixtures on plasma-treated microstructured surfaces as varying with ethanol concentration.

Because the square-cusp microstructure primarily governs multidirectional spreading of the precursor film and has unobvious influence on the bulk droplet body, we used the cross-cusp microstructure—capable of programming bulk multidirectional spreading—to study how liquids with different γ (and slightly different η) map onto spreading modes and distances. The optical images in **Figure R2** capture the overall

trends, in which yellow circles mark the initial droplet deposition positions. Overall, when the ethanol concentration is 60% or below, droplets can spread multidirectionally along the predefined orientations, and the spreading length increases with increasing concentration. At 80% concentration, due to lower surface tension and enhanced wettability, the droplet body still shows a multidirectional trend but with weakened performance. In this case, slight unintended spreading occurs along the pinning direction. It seems that the cusps cannot stop the precursor film for droplet with low surface tension, thus disabling the programmed multidirectional spreading. For pure ethanol, the droplet completely loses its multidirectional spreading capability and exhibits isotropic wetting. These observations indicate that surface tension is an important factor influencing both the mode and stability of multidirectional spreading, with low concentration values (0–60% ethanol) favoring controllable and stable performance can be achieved, while excessively low surface tension weakens or disrupts this ability.

Figure R2. Spreading behaviors of ethanol–water mixtures with different

concentrations on plasma-treated microstructured surfaces under 5 spreading modes (C-pinning, C-mode I–IV).

To further quantify the spreading behaviors under different modes, we analyzed the dimensionless spreading lengths, defined as the ratio of spreading length to the initial droplet radius, along the four characteristic directions (x^- , x^+ , y^- , y^+) for ethanol–water mixtures of varying concentrations. The quantitative analysis of dimensionless spreading length in **Figure R3** shows a general trend consistent with the qualitative patterns observed in **Figure R2**. Specifically, the desired guidance directions exhibit long spreading lengths at low ethanol concentrations, followed by a decline at higher concentrations due to the loss of directional confinement, whereas the pinned directions remain suppressed until ethanol content is sufficiently high to weaken surface tension and induce leakage. In the C-pinning state, all four directions almost show uniform spreading length for mixtures with low ethanol concentration. The spreading lengths gradually increase with concentration, reaching values of about 3 at 80%. For pure ethanol, the predefined pinning effect is completely lost, leading to long spreading lengths in all directions. In C-mode I, where y^+ is the desired guidance direction, the spreading length grows sharply from about 4.5 at 0% to nearly 8 at 40%, then decreases with further ethanol addition as stronger wetting broadens the droplet and reduces anisotropy. In C-mode II, the desired guidance directions x^+ and y^+ exhibit a clear increase-then-decrease trend, reaching maxima of about 4 around 40–60%, while the pinned direction remains well constrained below this concentration range. C-mode III follows a similar pattern: the desired guidance directions x^- , x^+ , and y^+ peak at about 4 around 40–60%, whereas the pinned direction maintains effective restriction. In C-mode IV, since all four axes are designated as guidance directions, the spreading lengths in all directions increase progressively with the increase of ethanol concentration, which is obvious different to other modes. Overall, these results demonstrate that low ethanol content ($\leq 60\%$) promotes long-range, anisotropic spreading in guided modes, whereas high ethanol content ($\geq 80\%$) implies excessive reduction of surface tension compromises pinning and leads to nearly isotropic spreading behavior.

Figure R4 presents the maximum spreading speeds of ethanol–deionized (DI) water mixtures droplets on different specimens, showing a consistent pattern in which speeds along the guided directions increase steadily with concentration and peak near 80 %, while the pinned directions remain very low spreading speed with only slight increases with the increase of ethanol. At 100 % ethanol the enhanced wettability removes anisotropy so speeds are high in all directions but slightly below those along the guided axes at 80 %. In the C-pinning mode, when ethanol concentration is ≤ 60 %, all four pinned directions stay slow ($<1.5 \text{ mm}\cdot\text{s}^{-1}$), then rise markedly as the ethanol concentration increases to 80% and 100 %, indicating weakened pinning. Across C-mode I–IV, as concentration increases from 0 % to 80 %, the desired guidance directions consistently sustain rapid motion, whereas the pinned directions remain slow with only minor growth. For example, in C-mode I the guided y^+ direction, the spreading speed reaches about $43 \text{ mm}\cdot\text{s}^{-1}$ at 80 %, while along the pinned axes, the droplet remains near baseline. Consequently, these results imply that for ethanol–deionized (DI) water mixtures with an ethanol concentration lower than 60 %, the droplet presents anisotropic spreading dominated by the guided directions. While, when the ethanol concentration is higher than 80 %, the ethanol sacrifices spreading directionality, thus presenting uniform spreading behaviors.

Figure R3. Dimensionless spreading lengths of droplets with different ethanol concentrations along x -, x +, y - and y + directions under 5 spreading modes (C-pinning, C-mode I-IV).

Figure R4. Spreading velocities of droplets with different ethanol concentrations along x -, x +, y - and y + directions under 5 spreading modes (C-pinning, C-mode I-IV).

The possible mechanism can be analyzed as follows. According to **Equation (7)** in the revised manuscript, when the contact angle θ equals to α (30°), the resistance force reaches its peak value. For pure water, the contact angle is 35° , which is rather close to α (30°). In this case, the pinning effect for precursor film is very strong. The driving force for precursor film in desired directions, however, is high. Hence, directional spreading of precursor film occurs along the expected direction, which drag the main droplet body for directional spreading. With the increase of ethanol concentration, the contact angle decreases, indicating the enhancement of driving force for precursor film in desired directions, as shown in **Equation (4)** in the revised manuscript. Meanwhile, the resistance force for precursor film in undesired directions is decreased according to **Equation (6)** in the revised manuscript. This may explain for ethanol-water mixtures with high ethanol concentration, the spreading of precursor film is more obvious, even spreading in undesired directions, such as 80 % and 100 %

ethanol-water mixtures. The precursor film in undesired direction may drag the main droplet body, resulting in spreading of the droplet in undesired guidance directions. Such effect may limit the droplet spreading in desired guidance directions since undesired spreading may reduce the flow rate. Consequently, although the decrease of contact angle implies the increase of wettability, spreading lengths of the main droplet body in desired directions may be shortened, as shown in ethanol-water mixture with 60 %, 80 % and 100 % in **Figure R2**.

To further examine the spreading behavior for liquids with surface tensions higher than that of water, we employed an equivalent approach by regulating the static contact angle on untreated silicon wafers. On untreated wafers, the static contact angle was measured to be 42.18° for 60 % ethanol solution, 60.3° for 40 % ethanol solution, and 95.5° for pure water. **Figure R5** presents the spreading behaviors of droplets with different concentrations on the 5 specimens having different spreading modes (C-pinning, C-mode I–IV). The results show that with small contact angles (e.g., 60 % ethanol, $CA=42.2^\circ$), droplets readily spread along multiple guided directions, exhibiting long distances and strong anisotropy. At intermediate wettability (40 % ethanol, $CA=60.3^\circ$), spreading remains multi-directional but becomes more moderate in range, with precursor films beginning to show reduced continuity in the guided directions. For pure water with the highest contact angle ($CA=95.5^\circ$), spreading is significantly suppressed, as both precursor films and droplet bodies lose directional spreading ability. In this state, droplets are pinned regardless of the surface microstructures (spreading modes), with negligible extension along any direction. These observations demonstrate that static contact angle can effectively serve as an equivalent parameter to explore high-surface-tension conditions, while also revealing that increasing contact angle progressively weakens precursor film spreading, shortens droplet extension along guided directions, and eventually eliminates anisotropy.

Figure R5. Spreading behaviors of ethanol–water mixtures with different static contact angles on untreated silicon surfaces under 5 spreading modes (C-pinning, C-mode I–IV).

To further evaluate the spreading behavior under different static contact angles, we quantitatively analyzed the dimensionless spreading length and the maximum spreading speed, as summarized in **Figure R6** and **Figure R7**. The results reveal clear overall trends. In terms of spreading length (**Figure R6**), liquids with low contact angles, i.e. the 60 % ethanol solution (42.2°), exhibit the longest spreading length along the guided directions, typically exceeding two to three times the initial droplet radius, while the pinned direction remains largely suppressed near unity. At intermediate contact angles (around 60.3°), the spreading length decreases slightly, reflecting weakened guidance. For pure water with relatively high contact angle (95.5°), directional spreading is almost completely inhibited, with dimensionless spreading lengths close to one regardless of mode.

As shown in **Figure R7**, the specimens with lower contact angles achieve fast spreading along guided directions, often above 10 mm/s, while pinned directions remain below 2 mm/s. At intermediate angles, speeds drop correspondingly. When $CA=95.5^\circ$, spreading lengths remain uniformly low across all directions, confirming that increased hydrophobicity eliminates both precursor film spreading and anisotropic extension.

Figure R6. Dimensionless spreading lengths of droplets with different static contact angles along x^- , x^+ , y^- and y^+ directions under 5 spreading modes (C-pinning and C-modes I-IV).

Figure R7. Spreading velocities of droplets with different static contact angles along x^- , x^+ , y^- and y^+ directions under 5 spreading modes (C-pinning, C-mode I-IV).

The possible mechanism can be analyzed as follows. For droplets with relatively

high contact angle, with the increase of contact angle, the wetting behaviors of the main droplet body become more unobvious, indicating shorter spreading length and lower spreading speed. This phenomenon is easy to be understood if we can imagine the lotus effect. For the precursor film, as shown in **Fig. 3c** and **Equation (4)** in the revised manuscript, with the increase of contact angle, the total driving force is decreased. Hence, in **Figure R5**, for droplets having high contact angle (i.e. 0 % and 40 %), the precursor film cannot be generated from the main droplet body. Without the drag effect precursor film, the spreading of main droplet body become weak, which also contributes to the insignificant spreading performance for droplets with high contact angle.

The above results mainly focused on the multidirectional spreading behavior of liquids with different surface tensions and static contact angles. However, since the viscosity differences among ethanol–water mixtures are relatively small, it is difficult to establish a clear correlation between viscosity and spreading behavior. To address this, we selected four representative liquids with comparable surface tensions in the range of 25–40 mN/m but with markedly different viscosities, namely 80% ethanol–DI water mixtures (80% EWM), white oil, paraffin oil, and castor oil. Their physical properties are summarized in **Figure R8**, which shows that despite the comparable surface tension values, their viscosities span a wide range. The static contact angles of all four liquids on both plasma-treated and naturally aged surfaces remain below 10°, ensuring equivalent wetting conditions for subsequent spreading tests and highlighting viscosity as the dominant variable.

Figure R8. Surface tension, dynamic viscosity and static contact angle of 4 different liquids (80% ethanol–DI water mixtures (80% EWM), white oil, paraffin oil, and castor oil). a Surface tension and dynamic viscosity of 4 liquids. b Contact angle of 4 liquids on plasma-treated microstructured surfaces.

To further investigate the effect of viscosity, multidirectional spreading experiments were conducted under the C-mode IV configuration. This mode was selected because viscous liquids are typically oils, which generally exhibit low surface tension and strong wettability. In the other four configurations, the presence of preset pinning directions would be invalidated by such strong wettability. **Figure R9** presents the time-sequence images of droplet spreading for the four liquids with different viscosities. The results reveal that for low-viscosity liquids (e.g., 80% ethanol–DI water mixtures), spreading is rapid and reaches equilibrium within a few seconds, showing smooth and extensive precursor film propagation. For liquids with relatively high viscosity (white oil and paraffin oil), the spreading becomes slower and the final coverage decreases. For the highest-viscosity liquid (castor oil), spreading is significantly retarded, with the droplet remaining nearly pinned even after 10 s, demonstrating that viscosity strongly suppresses dynamic spreading despite favorable wettability. These results confirm that while surface tension determines the driving force, viscosity governs the speed of multidirectional spreading.

Figure R9. Spreading behaviors of 80% ethanol–DI water mixtures (80% EWM), white oil, paraffin oil, and castor oil.

To systematically evaluate how viscosity influences multidirectional spreading, we performed a quantitative analysis of dimensionless spreading length and maximum spreading speed for four representative liquids of comparable surface tension but distinctly different viscosities, as illustrated in **Figure R10**. In **Figure R10a**, the dimensionless spreading length shows only a modest dependence on viscosity. Although a slight downward trend is observed as viscosity increases, the differences among 80% ethanol solution, white oil, and paraffin oil remain relatively small, all maintaining values above 2 along the guided directions. Only castor oil, with the highest viscosity of about 650 mPa·s, exhibits a significantly reduced spreading length close to 2, indicating that extremely high viscosity begins to constrain the liquid spreading. This comparison suggests that viscosity is not the primary factor governing spreading length, as the microstructured surface ensures a comparable level of directional transport across low- and medium-viscosity liquids. In contrast, **Figure R10b** highlights that viscosity exerts a much stronger influence on spreading speed. The 80% ethanol solution, characterized by low viscosity, achieves the fastest motion with peak velocities of approximately 20 mm/s, whereas white oil and paraffin oil show moderate reductions to around 12–13 mm/s and 8–9 mm/s, respectively. Castor oil, due to its very high viscosity, spreads extremely slowly, with maximum speeds of only ~2.5 mm/s, reflecting the dominant viscous resistance that suppresses dynamic wetting. Taken together, these results reveal that viscosity plays a limited role in determining spreading length but acts as a decisive parameter in controlling spreading speed.

Figure R10. Quantitative analysis of multidirectional spreading for 80% ethanol–DI water mixtures (80% EWM), white oil, paraffin oil, and castor oil with different viscosities. a Dimensionless spreading length of four liquids with different viscosities. **b** Spreading velocities of 4 liquids along x^- , x^+ , y^- and y^+ directions under C-modes IV.

The conclusions drawn from the above experiments can be also supported by our previous study (See Ref. 1 in the response). The spreading speed of oil with low viscosities is obviously lower than that of high viscosity oils.

Figure R11. Snapshots of unidirectional spreading for castor oil, PAO6, and PAO4 with different viscosities¹.

We sincerely thank the reviewer once again for this constructive suggestion, which has greatly strengthened the universality and robustness of our design. The additional experiments on liquids with varied surface tensions and viscosities provide a more comprehensive validation of the proposed criteria and directional spreading behaviors.

Relevant analyses, figures, and data have been incorporated into the revised manuscript (**Section 2.2 in Pages 8–9**) and Supporting Information (**Supplementary Figs. S18–S27**), with all modifications clearly marked in red for ease of review.

2. Spreading velocity is a critical parameter for many practical applications. The manuscript lacks quantitative results on spreading dynamics, which are essential for evaluating performance. The authors should report and discuss the spreading velocity.

Response 2: We sincerely thank the respective reviewer for emphasizing the importance of reporting spreading velocity as a quantitative parameter. This point is crucial, as spreading velocity not only reflects the dynamic process of droplet wetting but also provides a direct measure to evaluate the performance of the proposed surfaces. We fully agree that including spreading velocity analysis significantly strengthens the manuscript.

To address this, we analyzed the spreading dynamics of droplet bodies on cross-cusp and square-cusp microstructured surfaces. **Figure R12** and **Figure R13** show the normalized spreading length over time for droplets along the four principal directions, together with their corresponding maximum spreading velocities. Across all configurations, a common feature is that the normalized spreading length rises rapidly in the initial stage and then reaches a plateau, while the peak spreading velocity appears within this rapid-growth phase. Importantly, both the spreading length and the spreading velocity exhibit strong consistency with the preset orientation of the microstructures, confirming the directional guidance effect. For the cross-cusp surfaces, C-mode I shows anisotropic dynamics, with the droplet front along the y^+ axis reaching a maximum velocity of ~ 26 mm/s, while the other directions remain below 2 mm/s. In C-mode II, the main spreading occurs along the x^+ and y^+ axes, with peak velocities of ~ 20 mm/s, while the speed velocities keep below 2 mm/s in the opposite directions. C-mode III and C-mode IV also exhibit clear alignment between spreading behavior and the designed orientations, with maximum velocities in the range of ~ 13 – 16 mm/s,

demonstrating that all cross-cusp modes induce preferential spreading consistent with their geometrical layouts. Another phenomenon should be noticed is for microstructures with less desired guidance directions have higher spreading velocities than those of microstructures with more desired directions. For example, C-mode I has a maximum speed of 26.45 mm/s, while C-mode III has a maximum speed of 20.79 mm/s. The possible reason for this phenomenon may be microstructures with less desired direction implies less leakage outlets, indicating higher flow rate for spreading.

Figure R12. Time-dependent spreading of the droplet body on cross-cusp microstructure under various spreading modes.

Figure R13. Spreading velocities of the droplet body on cross-cusp microstructure

under various spreading modes.

For the square-cusp surfaces (**Figure R14** and **Figure R15**), the normalized spreading length likewise shows directional dependence, with spreading lengths and velocities aligned with the structural orientation. However, the overall spreading performance is weaker compared to cross-cusp surfaces. In S-mode I, the y^+ axis reaches a maximum velocity of ~ 6 mm/s, while other directions remain ~ 1 – 2 mm/s. S-mode II shows velocity of ~ 4 – 5 mm/s along x^+ and y^+ , and S-mode III and IV yield more balanced but slower dynamics across directions, with peak velocities remaining in the 3 – 5 mm/s range. Thus, both cross-cusp and square-cusp micro-structured surface types establish directional consistency between spreading length and velocity. While, comparing with the cross-cusp structure type, the square-cusp structures generally yield lower velocities and shorter spreading lengths.

Figure R14. Time-dependent spreading of the droplet body on square-cusp microstructure under various spreading modes.

Figure R15. Spreading velocities of the droplet body on square-cusp microstructure under various spreading modes.

A direct comparison of cross-cusp and square-cusp surfaces (**Figure R16**) highlights both the maximum spreading velocity and the total spreading velocity summed over the four principal directions. The cross-cusp surfaces exhibit markedly higher values, with maximum velocities along the preferred axes reaching up to ~ 26 mm/s, and the combined contributions from all four directions significantly exceeding those of the square-cusp surfaces. In contrast, the square-cusp surfaces remain below ~ 6 mm/s in maximum velocity, and their total directional sum is much smaller. This comparison shows that while both geometries maintain consistency between spreading velocity and structural orientation, the cross-cusp surfaces not only enable faster motion along dominant directions but also achieve a higher overall spreading capacity. The square-cusp surfaces, although slower, distribute spreading more evenly among multiple directions, reflecting a weaker but still structurally guided dynamic.

Figure R16. Comparison of spreading velocities of droplets along four principal directions for cross-cusp and square-cusp surfaces.

We once again thank the reviewer for this constructive suggestion. In the revised manuscript (Section 2.2 in Page 6 and Fig. 2g in Page 7) and Supporting Information (Supplementary Figs. S10–S12), we have included quantitative spreading velocity results and related discussion, with all new content clearly highlighted in red for the reviewer’s convenience.

3. The authors propose applications in lubrication and thermal management. However, comparing their surface with a bare (non-structured) surface is insufficient. To convincingly demonstrate the contribution of directional spreading, a proper control—such as a surface with similar microstructures but without cusp features—should be used.

Response 3: We sincerely thank the respected reviewer for this insightful comment. We fully agree that comparing only with a bare, unstructured surface does not provide a comprehensive understanding of the contribution of cusp features, and that incorporating a proper control surface is essential for convincingly demonstrating the role of directional spreading.

To address this concern, we additionally fabricated and tested a surface with cross microstructures without cusp, in comparison with the bare surface and the cross-cusp microstructured surface (Figure R17). Tribological experiments were conducted under water-lubricated conditions using a reciprocating pin-on-disk tribometer (WDT-1, Shandong Huaqi Instrument Co., Ltd., China) with a 45# steel disk (surface roughness $\sim 0.5\ \mu\text{m}$, hardness HRC48) as the lower friction pair and a 6 mm copper pin (surface roughness $\sim 0.5\ \mu\text{m}$, hardness HRC30) as the upper friction pair. The load was set to 30 N and the sliding frequency to 4 Hz, each cycle covering a sliding distance of 12 mm. Before testing, $\sim 50\ \mu\text{L}$ of deionized water was applied to the contact zone, and all tests were repeated three times for reproducibility.

Figure R17. Optical images of three types of surfaces including bare surface, cross microstructured surface, and cross-cusp microstructured surface.

The results are shown in **Figure R18**. Under 30 N and 4 Hz (**Figure R18a**), the bare surface exhibited a friction coefficient rising from ~ 0.16 to ~ 0.30 with strong fluctuations, while the cross surface without cusp tips showed similar behavior, stabilizing around ~ 0.28 – 0.30 . In contrast, the cross-cusp surface maintained a much lower and steadier coefficient of ~ 0.20 . Across a wide load range of 10–50 N (**Figure R18b**) and frequency range of 2–6 Hz (**Figure R18c**), the cross-cusp surface consistently showed the lowest friction, with reductions of ~ 15 – 35% compared to the bare case and ~ 10 – 25% compared to the cross without cusps. At low loads, the friction coefficient reached as low as ~ 0.13 , demonstrating robustness over diverse conditions. The wear morphologies further confirmed these findings (**Figure R19**). Severe grooves and tearing were observed on the bare surface, and the cross microstructured surface also showed obvious scratches and partial wear. In contrast, the cross-cusp microstructured surface largely retained a smooth and intact profile, with only shallow scars.

These results clearly demonstrate that the cusps are indispensable for directional spreading and lubrication improvement. While cross microstructures alone provide little advantage compared to a bare surface, the incorporation of cusp tips establishes capillary-driven spreading pathways that continuously replenish lubricant into the contact region, prevent backflow, and stabilize the liquid film. This structural mechanism explains why only the cross-cusp surface achieved markedly reduced friction, suppressed wear, and robust lubrication performance, confirming the functional significance of cusp features.

Figure R18. Friction coefficients of three types of surfaces under different test conditions. a Friction coefficient curves under a load of 30 N and frequency of 4 Hz. **b** Average friction coefficient under different loads. **c** Average friction coefficient under different frequencies.

Figure R19. Post-test wear morphologies of bare surface, cross microstructured surface and cross-cusp microstructured surface after tribological experiments.

To complement the application part, we further carried out evaporation-based cooling experiments to evaluate the thermal management performance of different surfaces. The tests were performed by placing the prepared silicon wafers on a heating plate maintained at 50 °C, while an infrared thermal camera (HM-TPK20-3AQF/W,

HIKMICRO, China) was used to continuously record the surface temperature evolution. A $\sim 5 \mu\text{L}$ droplet of deionized water was dispensed with a micropipette for single-drop evaporation, and for continuous evaporation experiments, identical droplets were added every 30 s. All experiments were conducted in still-air laboratory conditions to ensure reproducibility and avoid external airflow effects. Bare wafers, square microstructures, and square-cusp microstructures were observed using an optical microscope, as illustrated in **Figure R20**.

Figure R20. Optical images of bare surface, square microstructured surface, and square-cusp microstructured surface.

The results of the single-drop evaporation experiments (**Figure R21** and **Figure R23a**) highlight the distinct behaviors of the three surfaces. On the bare surface, cooling was restricted to the droplet footprint, with the average temperature only decreasing from $\sim 49 \text{ }^\circ\text{C}$ to $\sim 42 \text{ }^\circ\text{C}$ within 100 s. The square-microstructured surface exhibited partial enhancement, reducing the temperature to $\sim 36 \text{ }^\circ\text{C}$ in the same period, though the cooling distribution remained non-uniform. In contrast, the square-cusp microstructure enabled both rapid and spatially uniform cooling, with the average surface temperature decreasing from $\sim 49 \text{ }^\circ\text{C}$ to $\sim 27 \text{ }^\circ\text{C}$ in just 60 s. The corresponding thermal maps confirmed that the entire surface was effectively wetted, demonstrating superior evaporation efficiency.

The repeated-droplet evaporation tests (**Figure R22** and **Figure R23b**) provided further insights into sustained thermal management. Under repeated injection over 480 s, the bare surface exhibited large thermal fluctuations and pronounced localized hot spots, while the square-microstructured surface provided somewhat smoother cooling but showed gradual deterioration over time. The square-cusp surface, however, maintained a stable cooling performance, with the average temperature consistently

stabilized in the narrow range of 24–26 °C. The temperature curves further confirm this stability, as fluctuations on the square-cusp surface remained minimal compared with the other two samples, highlighting its ability to provide long-lasting and sustainable cooling.

Figure R21. Time-sequenced infrared thermal images after a single 5 µL deionized water droplet deposition onto bare surface, cross microstructured surface and square-cusp microstructured surface.

Figure R22. Time-sequenced infrared thermal images under periodic deposition of 5 µL deionized water about every 30 s onto bare surface, square microstructured

surface and square-cusp microstructured surface.

Figure R23. Average surface temperature variations of bare surface, square microstructured surface and square-cusp microstructured surface over time. a Temperature evolution after a single 5 μL water droplet evaporation experiment. **b** Temperature evolution under repeated 5 μL water droplet deposition every 30 s.

The above findings underscore the critical role of cusp features in evaporation-driven cooling. The square-cusp microstructures promote the formation and guided spreading of precursor films, enlarging the effective wetting area and redistributing liquid across the surface. This process enables distributed and uniform evaporation, ensuring high efficiency and stability under both single and repeated droplet conditions. By combining rapid cooling with long-term stability, this strategy provides a structurally simple, scalable, and non-invasive approach for localized thermal management, with strong potential for advanced heat dissipation systems.

We are deeply grateful to the reviewer for this valuable comment, which prompted us to strengthen the work through more rigorous controls and extended analyses. All related additions and revisions have been incorporated into the revised manuscript (Section 2.4 and Section 2.5 in Pages 13–16) and Supporting Information (Supplementary Figs. S33–S34, Figs. S36–S37 and Movies 9–10), with the new content highlighted in red for clarity.

4. In Equation 6, the contribution of the bottom surface is not included. A justification

or correction is needed to ensure completeness and consistency in the physical modeling.

Response 4: We sincerely thank the reviewer for this insightful comment, which has helped us to improve the rigor and completeness of the physical modeling presented in our manuscript. Due to our oversight, the previous expression in Equation 6 did not explicitly reflect the contribution from the bottom interface. We have now carefully revised this section as follows.

Considering the complexity of force analysis on the pinned liquid interface, the liquid surface at the instant of pinning is regularized as follows: it is assumed that intermolecular tension within the liquid can be neglected, the pinned liquid surface is cylindrical, and the contact angle between the liquid and the hydrophilic sidewalls equals the intrinsic contact angle of the liquid on the material. Only the attractive forces between the liquid and the sidewalls as well as the bottom surface are taken into account. For the case where the inclination angle satisfies $\alpha \leq \theta < 90^\circ$, the total resistance force F' acting on the precursor film during reverse spreading can be decomposed into two components:

Specifically, the capillary resistance from the two inclined sidewalls is retained as:

$$F_b' = 2\gamma h \cos(\theta - \alpha) \quad (\text{R1})$$

In addition, we now introduce the contribution from the bottom surface, which arises from the horizontal component of surface tension along the substrate:

$$F_{d2}' = \gamma l \quad (\text{R2})$$

Accordingly, the total resistance force F' is expressed as:

$$F' = F_b' - F_{d2}' = 2\gamma h \cos(\theta - \alpha) - \gamma l \quad (\text{R3})$$

This revised formulation accounts for the capillary forces and more accurately describes the physical resistance that hinders the retreat of the precursor film in the reverse direction. Consequently, the pinning criterion in the backward direction is no longer based solely on the sidewall component F_b' , but is now determined by the total capillary resistance F' , offering a more comprehensive and physically grounded

description.

In alignment with this theoretical refinement, we have also updated the force analysis schematic and the total capillary resistance F' (Equation 6) for the backward precursor film. The revised diagram is now shown as **Figure R24** (replacing **Fig. 3(e)** in the manuscript), in which all capillary force contributions are clearly annotated with directional vectors. In addition, the original **Fig. S30b** in the Supporting Information has been updated accordingly and is now provided as **Figure R25**.

Figure R24. Schematic of the backward pinning force acting on the precursor film due to sharp cusp edges, including sidewall force F_b' and bottom force F_{d2}' .

Figure R25. Design space for backward pinning defined by the intrinsic contact angle θ and cusp-to-bulk angle α , where the shaded region satisfies the condition $F' > 0$ for effective precursor film pinning.

We again thank the reviewer for pointing out this important detail, which has led to a more complete and physically consistent model. Furthermore, the related descriptions and mathematical derivations in the Supporting Information have been

thoroughly revised to maintain consistency with the corrected model. All these modifications—including the revised diagram, updated equations, and explanatory content—have been clearly marked in red font in the revised manuscript (**Section 2.3 in Page 10 and Fig. 3e**) and Supporting Information (**Supplementary Note 3 and Fig. S30b**) to facilitate reviewer verification.

5. It is unclear whether directional spreading occurs under continuous water supply or only for a single droplet. The latter would significantly limit practical applications. The authors should clarify this and ideally provide experimental evidence under continuous conditions.

Response 5: We sincerely thank the respected reviewer for this important comment and fully agree that clarifying whether directional spreading persists under continuous liquid supply is essential for assessing practical applicability.

To address this, we conducted supplementary experiments using a microinjection pump (ZS100-01, Baoding Chuangrui Precision Pump Co., Ltd., China) to continuously inject water at a flow rate of 10 $\mu\text{L}/\text{min}$. The results are presented in snapshots in **Figure R26** (cross-cusp surface) and **Figure R27** (square-cusp surface). In both cross-cusp and square-cusp surfaces, the droplet body maintains clear directional extension along the predesigned axes, while the precursor film spreads slightly ahead but remains highly coordinated with the body, thereby sustaining the guided transport.

In **Figure R26**, corresponding to the cross-cusp surface, the four modes (C-mode I–IV) demonstrate that the droplet body consistently follows the anisotropic geometry, forming elongated or branched spreading fronts depending on the mode. Notably, the propagation is stepwise at cusp junctions, highlighting the strong pinning–depinning dynamics characteristic of this structure. In contrast, **Figure R27**, corresponding to the square-cusp surface, shows that in all four modes (S-mode I–IV), spreading proceeds more smoothly and rapidly along the imposed axes, reaching a steady footprint earlier compared to the cross-cusp case.

These results reveal both commonalities and differences. The commonality is that both structures enable the droplet body to achieve stable and guided spreading under continuous injection, with strong coupling between the precursor film and the droplet body. The differences lie in the dynamics—cross-cusp surfaces induce more stepwise, pinning-dominated progression with enhanced confinement, while square-cusp surfaces support faster and more uniform spreading. These complementary characteristics underscore the versatility of the designed geometries in regulating liquid transport under practical continuous-flow conditions.

All clarifications, experimental results, and corresponding figures have been added to the revised manuscript (Section 2.2 in Page 8 and Fig. 2k in Page 7) and Supporting Information (Supplementary Figs. S16–S17 and Movies 3–4), with new content clearly marked in red for the reviewer’s convenience. We again thank the reviewer for this valuable suggestion, which has helped us reinforce the robustness of our findings.

Figure R26. Droplet spreading on cross-cusp surfaces under continuous injection

in 4 representative modes (C-mode I–IV).

Figure R27. Droplet spreading on square-cusp surfaces under continuous injection in 4 representative modes (S-mode I–IV).

Minor Comments:

1. The term “fluid spreading” should be revised to “liquid spreading” throughout the manuscript, as the study focuses specifically on liquids.

Response 1: We sincerely appreciate the reviewer’s insightful comment. We fully agree that “liquid spreading” is a more accurate term than “fluid spreading” in the context of our study. Although the word “fluid” can broadly refer to both liquids and gases flowing in confined spaces, our research exclusively involves the spontaneous capillary-driven transport of incompressible Newtonian droplets (specifically, deionized water) on micro-structured solid surfaces. There is no involvement of gaseous phases or complex multiphase liquids flowing in confined spaces, and no generalization beyond the liquid state is intended. In addition, fluid generally indicates

continuous flow of liquid and/or gas, which is not consist with the situation in our manuscript. Hence, liquid is more suitable for this manuscript.

Therefore, to ensure terminological precision and avoid ambiguity, we have carefully replaced all instances of “fluid spreading” with “liquid spreading” **throughout both the manuscript and the Supporting Information**. This revision ensures consistency between the terminology used and the actual scope of our experimental system.

For transparency and ease of review, all revised terms have been highlighted in red font in the revised manuscript and Supporting Information.

2. On Page 2, line 40: The statement “liquid spreading from high to low surface energy regions” is incorrect. In fact, liquids spread from regions of low to high surface energy. This should be corrected.

Response 2: We sincerely thank the reviewer for pointing out this important error. The original sentence stating that “liquid spreading from high to low surface energy regions” was indeed a result of an inadvertent wording mistake, and we fully agree that it contradicts the fundamental principles of interface science.

In reality, liquids tend to spread spontaneously from low to high surface energy regions. This is because higher surface energy solids promote better wetting, corresponding to lower equilibrium contact angles. According to Young’s equation, the contact angle decreases as the solid–liquid interfacial energy decreases relative to the solid–vapor energy. Therefore, a liquid droplet on a surface will preferentially move toward areas of higher surface energy, where the interfacial free energy can be minimized through increased wetting.

We have corrected the statement on Page 2, line 40 to: “Wettability gradients propel liquids spreading from low to high surface energy regions, but they often suffer from limited uniformity and unstable spreading performance.” This revision has been clearly marked in red font in the updated manuscript (**Section 1 in Page 2**).

Reviewer #2:

In this article, a novel type of microstructured surface, fabricated using standard photolithography, is proposed to achieve controllable, multidirectional fluid wetting.

The authors experimentally demonstrate that the proposed three-dimensional microstructures manipulate capillary forces to guide liquid spreading along specific directions without any external energy input.

I find the topic exciting, and both the research work and methodology are robust. This contribution advances our understanding of next-generation smart surfaces for a wide range of applications.

I recommend the publication of this article in Nature Communications, with only a few minor comments the authors may wish to consider:

1. Do the square-cusp microstructures have the same depth as the cross-cusp ones?

This is not mentioned in the article.

Response 1: We sincerely thank the respected reviewer for this question. Due to our oversight, the depth of the square-cusp microstructures was not explicitly described in the original manuscript.

To address this, we have now added laser confocal microscope (DCM8, Leica, Germany) characterization of the square-cusp microstructures to the revised Supporting Information, and also included it below as **Figure R28** for reference. Given the variety of structural modes, we selected the representative S-mode IV configuration for depth analysis. The results show that the square-cusp microstructures exhibit a uniform depth of approximately 12 μm , which is consistent with that of the cross-cusp microstructures.

We would also like to clarify that both the square-cusp and cross-cusp microstructures were fabricated in the same photolithography batch using identical processing parameters, including photoresist thickness, exposure time, and development conditions. Therefore, the measured depth of the S-mode IV structure reliably reflects the depth across all square-cusp variants.

This additional information and figure have been included in the revised Supporting Information (**Supplementary Fig. S3**), with the corresponding changes

marked in red font for clarity.

Figure R28. Three-dimensional morphology and cross-sectional profile of cross-cusp and square-cusp microstructures obtained by laser confocal microscope. a 3D topographic reconstruction of the cross-cusp microstructure (C-mode IV), highlighting the cross-shaped body with cusps, while the cross-sectional profile indicates a uniform structural depth of approximately 12 μm . **b** 3D topographic reconstruction of the square-cusp microstructure (S-mode IV), featuring the square-shaped body with cusps, while the cross-sectional profile reveals a consistent structural depth of approximately 12 μm .

2. Labels should be added to all panels of Figures S8 and S9 to indicate the corresponding mode.

Response 2: We sincerely thank the respected reviewer for this helpful suggestion. We agree that adding explicit labels to all panels of Figures S8 and S9 is necessary to clearly indicate the corresponding spreading modes and to improve readability and interpretation of the results.

In response, we have now added appropriate labels to each panel in Figures S8 and S9, denoting the specific mode corresponding to the droplet spreading behavior. These labels were carefully placed to avoid overlapping with key visual features, thereby ensuring clarity. We believe that this modification not only makes the figures easier to interpret at a glance, but also helps the reader to follow the logical progression of mode

evolution more intuitively.

The updated figures are also shown below as **Figure R29** and **Figure R30**, respectively, and have been incorporated into the revised Supporting Information (**Supplementary Figs. S10–S11**). All modifications have been marked in red font for clarity.

Figure R29. Time-dependent normalized spreading lengths (L/R) of the droplet body along eight directions on cross-cusp microstructures under various spreading modes.

Figure R30. Time-dependent normalized spreading lengths (L/R) of the droplet body along eight directions on square-cusp microstructures under various spreading modes.

3. In Equation 4, the term $\cos\theta$ appears on both sides of the equation. It would be clearer to solve the equation with respect to $\cos\theta$, so it appears only on the right-hand side.

Response 3: We sincerely thank the respected reviewer for the thoughtful and constructive suggestion. We fully agree that the original form of Equation (4), where the term $\cos\theta$ appears on both sides of the inequality, may hinder intuitive understanding and introduce unnecessary algebraic ambiguity. Rearranging the expression to isolate $\cos\theta$ on one side improves both clarity and usability, especially when applying the equation to evaluate design criteria or compare experimental parameters.

In response, we have now algebraically transformed Equation (4) so that $\cos\theta$ appears only on the left-hand side. The detailed rearrangement process is as follows:

Original form of Equation (4):

$$\cos\theta > \frac{2h}{l} [\sin(\alpha - \beta) - \cos\theta \cos(\alpha - \beta)] \quad (\text{R4})$$

Step 1: Expand the right-hand side:

$$\cos\theta > \frac{2h}{l} \sin(\alpha - \beta) - \frac{2h}{l} \cos\theta \cos(\alpha - \beta) \quad (\text{R5})$$

Step 2: Bring all $\cos\theta$ terms to the left-hand side:

$$\cos\theta + \frac{2h}{l} \cos\theta \cos(\alpha - \beta) > \frac{2h}{l} \sin(\alpha - \beta) \quad (\text{R6})$$

Step 3: Factor out $\cos\theta$ on the left-hand side:

$$\cos\theta \left[1 + \frac{2h}{l} \cos(\alpha - \beta) \right] > \frac{2h}{l} \sin(\alpha - \beta) \quad (\text{R7})$$

Step 4: Solve for $\cos\theta$:

$$\cos\theta > \frac{2h \sin(\alpha - \beta)}{l + 2h \cos(\alpha - \beta)} \quad (\text{R8})$$

This final form, now denoted as Equation (R8), expresses the minimum required $\cos\theta$ value as a function of geometric parameters and surface tension considerations, and avoids recursive dependence. It is especially useful for visualizing the feasible design space in **Fig. 3d**, where θ , α , and β are the key variables.

We have updated the revised manuscript (**Inequation (5) of Section 2.3 in Page 10**) to replace the original Equation (4) with this new expression, and the change is clearly marked in red font for ease of reference.

4. Since it is suggested that these microstructures could be used in advanced tribological systems, it would be useful to comment on their durability. In such applications, how would these surfaces behave in the presence of debris accumulation? Complex surface features might trap particles and accelerate wear.

Response 4: We sincerely thank the respected reviewer for this constructive comment and fully agree that durability is a critical factor to assess if these microstructured surfaces are to be applied in advanced tribological systems. As correctly noted, the accumulation of debris may compromise surface performance over long-term operation, and it is therefore necessary to evaluate wear resistance and its impact on functionality.

To address this, we conducted tribological durability experiments under well-controlled conditions. Specifically, a reciprocating tribometer (WDT-1, Shandong Huaqi Instrument Co., Ltd, China) was employed, with a normal load of 30 N, frequency of 4 Hz, stroke length of 6 mm (12 mm per cycle), and a total duration of 60 minutes (86400 cycles, corresponding to a total sliding distance of 10368 m). A 45# steel disk was used as the lower friction pair, into which four grooves were machined to embed the silicon wafers containing cross-cusp microstructures, bonded by PDMS to ensure surface levelness with the surrounding region. A copper pin with 6 mm diameter served as the upper counterpart. Prior to testing, both surfaces were polished and cleaned. Deionized water ($\sim 100 \mu\text{L}$) was supplied as the lubricant at identical positions for the structured surfaces. The coefficient of friction (COF) was continuously

recorded during the test.

The initial surface is shown in **Figure R31**, where (a) and (b) display optical images of the C-mode I and C-mode II cross-cusp microstructures prior to testing, without any debris. After 60 minutes of reciprocating sliding, **Figure R32a–b** illustrates that some debris accumulation indeed occurred on the microstructured surfaces, but the amount was relatively minor. Correspondingly, the friction coefficient curve (**Figure R32c**) demonstrates that although COF gradually increased from ~ 0.18 to ~ 0.20 over time, the overall rise was about 10 %, indicating that debris did not significantly deteriorate lubrication performance within this timescale.

These results can be explained as follows: i) the designed microstructures are located in the non-contact zones of the tribo-pair, meaning they are not directly subjected to wear, which limits debris entrapment; ii) the directional guiding capability of the microstructures not only channels lubricant toward the contact region but also inhibits the backflow of debris-laden lubricant from the contact area into the non-contact zone, thereby reducing the likelihood of debris accumulation within the structures. Such functionality is particularly advantageous under starved lubrication conditions, where maximizing lubricant utilization is crucial for reducing friction and wear.

Nevertheless, we acknowledge the reviewer’s point that under much harsher or prolonged conditions, significant wear in the contact zone could eventually lead to increased debris levels that may accumulate within the microstructures and impair their performance. Investigating this extreme case will be a meaningful direction for future work.

Figure R31. Optical microscopy images of cross-cusp microstructured surfaces before tribological testing. a C-mode I microstructure surface. b C-mode II

microstructure surface.

Figure R32. Tribological testing results of cross-cusp microstructured surfaces. a Optical microscopy image of C-mode I surface after testing. **b** Optical microscopy image of C-mode II surface after testing. **c** Friction coefficient curve.

We have incorporated the corresponding modifications and explanatory content into the revised manuscript (**Section 2.4 in Page 13**), with all new additions highlighted in red for clarity. We again thank the respected reviewer for raising this important point, which has allowed us to strengthen the work by addressing the durability considerations of the proposed surfaces.

5. Particles may also originate from dusty or dirty environments. How would the presence of such particles affect the efficiency of the proposed microstructures in general?

Response 5: We sincerely thank the respected reviewer for this insightful

comment. As the reviewer rightly noted, in addition to potential wear debris, particles originating from dusty or dirty environments could also influence the efficiency of microstructured surfaces. This is indeed an important concern when considering practical tribological applications.

In our work, all experiments were conducted in a cleanroom environment, which minimizes interference from environmental dust or contamination. However, to specifically assess the potential impact of dust contamination, we performed comparative tribological experiments under the same operating conditions as described in Comment 4 (applied load of 30 N, frequency of 4 Hz, sliding length of 6 mm, and test duration of 60 min). As a control, clean cross-cusp microstructured surfaces were examined, as shown in **Figure R31**, where no external particles were present. For the dust-contaminated case, we exposed otherwise identical microstructured surfaces to an open laboratory environment for one month, which resulted in noticeable dust particle adhesion on the surface (**Figure R33a**). Under the same test conditions, we then conducted reciprocating tribological experiments and monitored the coefficient of friction. The results demonstrate that the dust-contaminated surfaces exhibited friction coefficients within the same range as the clean surfaces (approximately 0.18–0.20), as shown in **Figure R33b**. While the contaminated surfaces displayed slightly larger fluctuations, the overall effect of dust on frictional performance was relatively small. These results suggest that limited particle adhesion does not significantly impair the functionality of the proposed surfaces.

It is important to emphasize that small particle contamination in real-world environments is unavoidable. To mitigate this effect, we recommend simple pre-cleaning procedures, such as short ultrasonic treatment, to remove adherent particles before use. Furthermore, in typical tribological applications, our designed microstructures are located in non-contact regions and often housed in enclosed systems, which reduces their direct exposure to dust and dirt.

Figure R33. Tribological comparison of cross-cusp microstructured surfaces under clean and dust-contaminated conditions. a Optical image of the dust-contaminated surface after one month exposure in laboratory environment. **b** Friction coefficient curves of clean and dust-contaminated surfaces under identical test conditions.

We fully acknowledge the respected reviewer’s concern that, under more severe or long-term conditions, excessive particle accumulation could eventually interfere with microstructure efficiency. However, we believe such effects can be alleviated through environmental control and periodic surface cleaning.

We have incorporated relevant explanations and additional information into the revised manuscript (**Section 2.4 in Page 13**), with all new content highlighted in red for the reviewer’s convenience. Once again, we are grateful to the respected reviewer for raising this important point, which has allowed us to broaden the scope of our discussion.

6. The article mentions that the intrinsic contact angle of the surface is 35°. How sensitive is the behavior of the proposed microstructures to surface wettability? For example, does their efficiency change significantly when using liquids with different surface tensions?

Response 6: We sincerely thank the respected reviewer for raising this important point. This comment is closely related to the concern highlighted by **Reviewer #1 in Major Comments 1** (Page 1 in *Response to Comments*), which emphasized the influence of liquid properties on spreading behavior. Indeed, the sensitivity of our proposed microstructures to surface wettability and their performance under liquids of varying surface tensions is a key issue for demonstrating the universality of the design. We fully agree with the respected reviewer that clarifying this aspect is essential for a comprehensive understanding of the robustness of our approach.

To investigate how liquid surface tension influences directional spreading, we prepared ethanol–deionized (DI) water mixtures with volume fractions of 0% (pure DI water), 20%, 40%, 60%, 80%, and 100% (pure ethanol), and examined their spreading behaviors on uniformly plasma treated microstructured surfaces. All substrates were plasma treated (Pluto-T, PLUTOVAC, Shanghai, China) at 100 W for 10 s, converting the initially hydrophobic surface (static contact angle $\approx 95^\circ$ before treatment) into a hydrophilic one. After treatment, the samples were stored in ambient air for 48 h prior to testing. As shown in **Figure R34**, surface tension decreases significantly with increasing ethanol concentration, from about 72 mN m^{-1} for pure DI water to about 22 mN m^{-1} for pure ethanol. Correspondingly, the static contact angle on the treated surfaces decreases monotonically, from $\sim 35^\circ$ for DI water to nearly 0° for ethanol. These results indicate that surface tension plays a central role in regulating the wetting performance of liquid.

Figure R34. Variation of surface tension and wettability for liquids with different ethanol concentrations. a Surface tension of ethanol–water mixtures varying with

ethanol concentration. **b** Contact angle of ethanol–water mixtures on plasma-treated microstructured surfaces varying with ethanol concentration.

Since the square-cusp microstructure mainly regulates the multidirectional spreading of the precursor film and exerts limited control over the bulk droplet, we employed the cross-cusp microstructure—which can program the multidirectional spreading of the droplet body—to examine how liquids with different surface tensions map onto spreading modes and distances. The optical images in **Figure R35** illustrate the overall behaviors. When the ethanol concentration is 60% or lower, droplets consistently spread multidirectionally along the predefined orientations, with spreading lengths increasing as concentration rises. When the ethanol concentration is 80%, reduced surface tension and stronger wettability allow the droplet body to maintain multidirectional spreading, but the performance becomes weaker: unintended spreading appears along the pinning direction, and the precursor film fails to sustain programmed spreading. For pure ethanol, droplets lose directional control entirely and display nearly isotropic wetting. These results qualitatively demonstrate that surface tension is a critical factor in determining both the mode and stability of multidirectional spreading, where low concentration values (0–60% ethanol) support controllable and robust performance, whereas excessively low surface tension undermines or disrupts this capability.

To gain deeper insight into the spreading performance of different modes, we examined the dimensionless spreading length, defined as the spreading length normalized by the initial droplet radius, along four characteristic directions (x^- , x^+ , y^- , y^+) for ethanol–water mixtures of different concentrations. The results in **Figure R36** are consistent with the qualitative features in **Figure R35**, while providing clearer mode-specific details. In general, the guided directions show a noticeable increase in spreading length with the increase of ethanol concentration at low ethanol levels, but this trend reverses at higher concentrations as directional confinement are lost, whereas the pinned directions remain constrained until surface tension becomes extremely low, leading to leakage. For the C-pinning case, liquid shape in all four directions initially

remain close to unity and then spreading lengths increase with the rise of concentration, reaching about 3 at concentration of 80%. With pure ethanol, pinning is entirely lost, and significant spreading occurs in every direction. In C-mode I, where the y^+ axis is the desired guidance direction, spreading increases sharply from around 4.5 at 0% ethanol to nearly 8 at 40%, followed by a decline at higher concentrations as stronger wetting reduces anisotropy. In this state, pure ethanol also eliminates single-direction guidance, while the other directions change only slightly. In C-mode II, the desired guidance directions x^+ and y^+ follow a clear rise-and-fall pattern, peaking at about 4 in the 40–60% range, whereas the pinned side remains constrained below this concentration. C-mode III shows a similar response, with desired guidance direction x^- , x^+ , and y^+ reaching about 4 in the same range, while the pinned direction shows only a small increase. In C-mode IV, since all 4 directions serve as guided orientations, the spreading lengths grow steadily with increasing ethanol concentration. Collectively, these findings indicate that low ethanol fractions ($\leq 60\%$) favor long-range and anisotropic spreading under guided modes, while excessive lowering of surface tension (ethanol concentration $\geq 80\%$) disrupts pinning and results in nearly isotropic spreading.

Figure R35. Spreading behaviors of ethanol–water mixtures with different

concentrations on plasma-treated microstructured surfaces under 5 spreading modes (C-pinning, C-mode I–IV).

Figure R36. Dimensionless spreading lengths of droplets with different ethanol concentrations along x -, x^+ , y - and y^+ directions under 5 spreading modes (C-pinning, C-mode I–IV).

Figure R37 summarizes the maximum spreading speeds of droplets with different ethanol concentrations under different modes. The results reveal a clear trend: speeds along guided directions rise steadily with increasing ethanol content and reach their maximum near 80 % or 100 %, while speeds in pinned directions stay low with only slight increases. For 100 % pure ethanol, stronger wettability eliminates anisotropy, leading to relatively high speeds in all directions. In the C-pinning case, when ethanol concentration is lower 60 %, all four pinned directions remain slow (lower than 1.5 mm s^{-1}), but speeds increase sharply when the ethanol concentration is higher than 80%. For C-modes I–IV, as the ethanol fraction grows from 0% to 80%, the desired guidance directions consistently sustain rapid motion of droplets, while pinned directions stay slow with only minor changes. For instance, in C-mode I the guided y^+ direction, the spreading speed reaches about $43 \text{ mm} \cdot \text{s}^{-1}$ at 80 %, while along the pinned axes, the

droplet remains near baseline. Overall, these results demonstrate that low ethanol content ($\leq 60\%$) promotes long-range, anisotropic spreading in guided modes, whereas high ethanol content ($\geq 80\%$) implies excessive reduction of surface tension compromises pinning and leads to nearly isotropic spreading behavior.

Figure R37. Spreading velocities of droplets with different concentrations along x-, x+, y- and y+ directions under 5 spreading modes (C-pinning, C-mode I–IV).

To further evaluate spreading behavior for liquids with surface tensions higher than water, we adopted an equivalent method by adjusting the static contact angle on untreated silicon wafers. On untreated wafers, the static contact angles were determined to be 42.2° for 60 % ethanol, 60.3° for 40 % ethanol, and 95.5° for pure water. **Figure R38** illustrates the spreading behaviors of droplets in these cases across five representative modes (C-pinning, C-mode I–IV). When the contact angle is relatively small, such as 42.2° for 60 % ethanol, droplets can easily spread along guided directions, achieving long spreading lengths with pronounced anisotropy. At an intermediate wettability, represented by 40 % ethanol with a contact angle of 60.3° , the spreading remains multidirectional, and the continuity of precursor films in the guided paths becomes less evident. In contrast, for pure water with high contact angle of 95.5° , spreading is strongly inhibited, as both precursor films and droplet bodies lose their ability to spread directionally. In this case, droplets are pinned in all modes with almost

no extension in any direction.

Figure R38. Spreading behaviors of ethanol–water mixtures with different static contact angles on untreated silicon surfaces under 5 spreading modes (C-pinning, C-mode I–IV).

To further investigate how static contact angle influences spreading, we carried out a quantitative assessment of the dimensionless spreading length and the maximum spreading speed, as summarized in **Figure R39** and **Figure R40**. The results display distinct and consistent patterns. Regarding spreading length (**Figure R39**), liquids with smaller contact angles—for example, the 60 % ethanol solution (42.2°)—present the farthest spreading along guided directions, typically reaching more than 2 to 3 times the original droplet radius, while the pinned direction remains effectively constrained near unity. At intermediate contact angles, such as 60.3°, the spreading length becomes limited, showing weaker guidance. At the highest contact angle of pure water (95.5°), directional spreading is almost entirely suppressed, with distances close to one in every mode, reflecting a pinned state.

For spreading speed (**Figure R40**), the overall behavior parallels that of distance. Droplets with small contact angles spread more rapidly along guided axes, often exceeding 10 mm/s, whereas the pinned directions remain below 2 mm/s. At intermediate angles, the speed decreases accordingly. When CA=95.5°, all directions exhibit uniformly low values, confirming that stronger hydrophobicity prevents both precursor film spreading and anisotropic transport. Overall, these results highlight that static contact angle determines not only the extent but also the speed rate of directional

spreading. Small angles enable efficient multidirectional spreading, whereas large angles result in pinned, isotropic behavior.

Figure R39. Dimensionless spreading lengths of droplets with different static contact angles along x^- , x^+ , y^- and y^+ directions under 5 spreading modes (C-pinning and C-modes I-IV).

Figure R40. Spreading velocities of droplets with different static contact angles along x^- , x^+ , y^- and y^+ directions under 5 spreading modes. (C-pinning and C-

modes I–IV)

We sincerely thank the reviewer again for this valuable comment, which has allowed us to clarify the role of static contact angle in governing both the length and the speed of directional spreading. This suggestion has strengthened the overall interpretation of our results and highlighted the broader significance of wettability in controlling liquid transport on structured surfaces. The corresponding revisions and supplementary analyses have been incorporated into the revised manuscript (**Section 2.2 in Pages 8–9**) and Supporting Information (**Supplementary Figs. S18–S24**), where they are marked in red for clarity.

Reviewer #3:

This paper presents an interesting investigation of bulk-cusp microstructures for controlling multi-directional fluid spreading, with potential applications in lubrication and thermal management. While the structures are very novel and the topic is of interest, the manuscript requires substantial improvements before it would be suitable for publication in a high-impact journal such as Nature Communications.

Major Concerns:

1. The manuscript claims a novel design, but the introduction cites numerous existing approaches to directional fluid control. The authors should consider making a stronger case for how the proposed bulk-cusp design represents a significant advance over existing microstructures. What specific limitations of prior work are overcome, and what new capabilities are demonstrated? A more thorough comparison to the existing literature is required, with quantitative metrics to support claims of improved performance.

Response 1: We sincerely thank the respected reviewer for this insightful comment. It rightly highlights the importance of positioning our proposed design in the context of existing research on directional fluid control. The suggestion to make a stronger case for how our bulk-cusp design surpasses prior approaches, and to clarify what specific limitations are overcome along with the new capabilities demonstrated, is highly valuable. We fully agree that such a comparison is essential for substantiating the novelty and significance of our work.

● Significant Advances of the Bulk-Cusp Design over Existing Microstructures

In response to the respected reviewer's comment, we emphasize that the proposed bulk-cusp microstructure constitutes a substantial advance over previously reported liquid-guiding structures. Its unique advantages can be summarized as follows:

- 1) **Functional breakthrough:** Most existing designs are limited to unidirectional liquid spreading, and only selective unidirectional spreading has been achieved in recent years²⁻⁴. Our bulk-cusp geometry fundamentally extends this capability to multidirectional control, enabling a simple structural design to achieve five distinct

liquid behaviors—pinning, unidirectional, bidirectional, tri-directional, and quad-directional spreading. This represents a significant functional expansion beyond prior art. Additionally, the mechanism for such multidirectional spreading is elucidated in depth in our study, which is helpful for enriching the knowledge of the influence of geometrical parameters on wetting behaviors.

- 2) **Structural simplicity:** Prior multidirectional structures often rely on complex three-dimensional architectures such as inclined pillars or microcavities, which are difficult to fabricate and scale⁴⁻⁶. By contrast, our design is realized through simple planar arrangements of cusp-shaped elements. This straightforward geometry not only eases fabrication but also facilitates large-area integration via standard lithographic processes.
- 3) **Broad applicability:** Previous reports on multidirectional transport mainly targeted different liquids, without clarifying how one liquid could be programmed into multiple spreading modes^{7,8,9}. Our design uniquely achieves this: for a single liquid, both bulk and precursor-film spreading can be tuned into different patterns. Furthermore, beyond deionized water, the structure enables stable multidirectional spreading of ethanol–water mixtures with surface tensions ranging from ~25 to 72 mN/m (up to 80% ethanol concentration). Such broad applicability underscores the robustness of the underlying mechanism.
- 4) **Pump-free and pump-assisted adaptability:** Achieving multidirectional spreading without external pumps or fields is extremely challenging, and has scarcely been demonstrated in prior studies¹⁰⁻¹⁵. Our design overcomes this fundamental limitation, allowing a single droplet to undergo rapid, pump-free multidirectional spreading. Moreover, it is fully compatible with continuous external pumping, enabling sustained delivery of liquid films where long-duration operation is required. This combination of difficulty-defying pump-free spreading and pump-assisted adaptability provides operational flexibility not previously reported.
- 5) **Application-level innovation:** The cross-cusp structures were innovatively

positioned in the non-contact regions of tribological pairs, enabling directional redistribution of lubricant and yielding a friction coefficient reduction of about 35%, which surpasses the performance of existing state-of-the-art surface strategies. Meanwhile, the square-cusp structures were applied to thermal management, where precursor-film spreading enabled both rapid temperature drops and localized, targeted cooling. Importantly, the bulk liquid continuously replenishes the precursor film, ensuring long-term evaporative cooling without external energy. These dual applications illustrate the wide-ranging impact and practical significance of the bulk-cusp microstructures for advanced scientific and engineering fields. For example, in the future, we can use this technology for flow-driven cell assembly.

- **Overcoming Prior Limitations and Demonstrating New Capabilities**

- 1) **Expanded directional capability**

Previous designs were largely restricted to unidirectional spreading, with only limited cases of selective unidirectional control. Our bulk-cusp design fundamentally breaks this limitation by achieving five distinct liquid behaviors—pinning, unidirectional, bidirectional, tri-directional, and quad-directional spreading—using a single architecture. This represents a leap from “one-direction control” to “programmable multidirectional control”, a capability not reported in prior studies.

- 2) **Fabrication-friendly architecture**

Earlier multidirectional microstructures often depended on complex 3D inclined pillars or cavities, which are difficult to fabricate and scale. Our design instead employs a planar cusp arrangement, fully compatible with standard lithography. This simplicity transforms what was once a specialized, small-scale approach into a scalable, integration-ready platform suitable for large-area applications.

- 3) **Programmable versatility**

Prior studies often tuned structures for different liquids but did not show how one liquid could be guided into multiple spreading modes. Our system uniquely demonstrates this: for a single liquid, both bulk and precursor-film spreading can be

programmed into diverse patterns. Moreover, it works across a broad liquid spectrum—from deionized water to ethanol–water mixtures with surface tensions between 25–72 mN/m (up to 80% ethanol), which can still achieve four-directional spreading. Such versatility underscores a mechanistic robustness that prior work lacked.

4) Breakthrough in pump-free spreading

Achieving multidirectional spreading without pumps or external fields has been a fundamental challenge in this field, and prior work scarcely demonstrated it. Our design not only achieves rapid pump-free multidirectional spreading from a single droplet but also remains compatible with external pumping, enabling long-duration operation with sustained precursor-film replenishment. This dual adaptability—difficult-to-achieve pump-free spreading combined with pump-assisted operation—marks a breakthrough beyond the state-of-the-art.

5) Quantified performance superiority

In tribological applications, state-of-the-art strategies generally reduce friction by <30%. Our cross-cusp structures achieved a ~35% reduction, surpassing these benchmarks. In thermal management, square-cusp structures sustained precursor-film replenishment, enabling localized cooling with rapid temperature drops without external energy, offering a decisive advantage for intelligent chip applications.

In summary, the two preceding parts establish both the conceptual advance and the concrete gap our bulk-cusp design closes relative to prior work. **Figure R41** benchmarks representative studies against five key axes—Multidirectional spreading, Structural simplicity, Liquid diversity, Self-driven capability, and Application validation—and shows that earlier approaches typically satisfy only a subset (e.g., unidirectional or selectively one-direction transport, 3D-complex architectures, pump/field-dependent actuation, or concept-only demonstrations). By contrast, This work traces the outer envelope on all five axes: (i) programmable multidirectional control (pinning → uni- → bi- → tri- → quad-directional), (ii) simple 2D cusp tessellations compatible with lithography, (iii) broad liquid diversity from water to ethanol–water mixtures (~25–72 mN m⁻¹), (iv) rapid self-driven spreading without

external pumps or fields while remaining pump-compatible, and (v) validated applications in lubrication ($\approx 35\%$ COF reduction via non-contact redistribution) and thermal management (sustained, directional precursor-film cooling for intelligent chip hotspots). This integrated comparison quantitatively clarifies why the bulk-cusp architecture constitutes a substantive step beyond state-of-the-art directional fluid microstructures.

Figure R41. Comparison of the proposed bulk-cusp design with previously reported directional liquid manipulation approaches^{2,3,6,12,15}.

● **Benchmarking Against Existing Literature with Quantitative Metrics**

Based on quantitative metrics to support the performance improvements, we provide a more comprehensive comparison with existing literature from the perspective of both lubrication and thermal management applications.

In our study, the cross-cusp microstructures positioned in the non-contact region created continuous capillary pathways that passively delivered lubricant into the interface. This mechanism maintained a stable friction coefficient of ~ 0.2 , while the bare surface rose to ~ 0.3 , yielding an overall reduction of about 35%. Such performance demonstrates that the proposed design can achieve substantial friction reduction without any external energy supply, while also ensuring long-term stability under dynamic conditions. In recent years, researchers have explored a variety of advanced strategies to reduce friction, which can generally be classified into four categories: protective or functional coatings, lubricant additives, surface texturing directly within the contact zone, and surface strengthening or modification techniques. Considerable

efforts have been devoted to each of these approaches, and their current status is summarized below to provide a clear benchmark for comparison.

Coating technologies have recently attracted considerable attention as an effective means of reducing friction. Jin et al. reported that a porous textured DLC/MAO multilayer coating markedly lowered the friction coefficient under dry and oil-lubricated conditions, with about 20.9% reduction achieved in the latter¹⁶. Lin et al. showed that chrome plating on polymer–metal friction pairs decreased the friction coefficient by around 20%, owing to improved hardness and the formation of a stable transfer film¹⁷. As a complementary path to coatings, lubricant additives provide a mature route to friction reduction with clear quantitative benchmarks. Dou et al. demonstrated that self-dispersed crumpled graphene balls in base oil lowered the friction coefficient by about 20% compared with pure oil, showing superior dispersion stability and tribological performance¹⁸. Hou et al. prepared graphene additives for low-sulfur diesel via plasma-assisted ball milling and reported a 20–24% reduction in the friction coefficient depending on the fuel type¹⁹. Karimi et al. showed that adding 0.2 wt% CuO nanoparticles to HB-80 turbine oil lowered the friction coefficient by 22.86%²⁰. Kumar et al. investigated h-BN nanoadditives in aerospace-grade lubricants and found that the addition of 0.1 wt% h-BN reduced the friction coefficient by 13% while enhancing lubricant stability²¹. Surface texturing of the contact zone has also been proven effective in lowering friction. Liu et al. investigated laser-induced circular dimples, and the textured surfaces exhibited up to a 27% lower friction coefficient compared with smooth counterparts²². In another study, Cao et al. examined cylindrical-ground microstructural surfaces and found that, under well-lubricated conditions, optimized textures improved friction performance by up to ~20% versus smoother, untextured counterparts²³. Chen et al. enhanced the tribological performance of cylinder guide bushes by introducing bionic convex textures with composite grease, resulting in a 5.0–6.7% reduction in friction coefficient²⁴. Ge et al. used multivariate linear regression to optimize micro-texture parameters and reported a 15.6% reduction in friction coefficient compared with texture-free surfaces²⁵. Beyond coatings, additives,

and in-contact texturing, other surface strengthening routes have also delivered measurable friction reduction. Cai et al. applied shot peening to CF53 steel and reported a 22.5% decrease in the friction coefficient relative to untreated samples²⁶. Zhang et al. strengthened GCr15 spherical joint bearings via ultrasonic surface rolling, achieving a 28% reduction in the friction coefficient²⁷. Hu et al. introduced low-frequency vibration during laser wire additive manufacturing of thin-walled Inconel 601, lowering friction coefficients by 12.6% (dry)–14.2% (oil)²⁸. Acharya et al. used surface mechanical attrition treatment on a Ti–Nb–Ta–O alloy and, in the $\beta+\alpha$ aged condition, observed a 21% reduction in friction coefficient under fretting²⁹.

From the above literature review, it can be concluded that the reduction rates achieved by currently advanced strategies—including coatings, lubricant additives, surface texturing, and other strengthening techniques—are consistently below 30% in the available reports. In contrast, as illustrated in **Figure R42**, our proposed method enables a friction coefficient reduction of about 35%, which is higher than those achieved by the existing approaches. The superior performance originates from the cross-cusp microstructures located outside the contact region, which passively and continuously replenish lubricant into the interface without the need for external energy. This design not only stabilizes the friction coefficient over time but also ensures long-term effectiveness under varying operating conditions. Therefore, the strategy is not merely an incremental improvement but represents an innovative route that couples structural design with passive capillary-driven transport, offering a novel and energy-efficient pathway for advanced lubrication management.

Figure R42. Comparison of friction coefficient reduction achieved by this work and representative advanced lubrication strategies.

In terms of thermal management, our work introduces a novel square-cusp microstructure that enables the directional spreading of a precursor film while leaving the droplet body almost stationary. This design ensures a continuous supply of precursor liquid that spreads outward in a sustained manner, thereby realizing efficient cooling of the heated substrate through evaporation. Such a mechanism represents a conceptual innovation with strong potential for applications in intelligent thermal regulation of electronic chips and other advanced devices. Importantly, it not only guarantees sustainable liquid supply but also offers programmable, directional, and intelligent control of heat dissipation, setting it apart from existing passive cooling approaches. At present, however, there are no directly comparable studies in the literature that provide quantitative benchmarks equivalent to our results. Instead, research in advanced thermal management has mainly progressed along several directions, such as surface modification to enhance evaporation, nanostructured coatings for improved heat transfer, capillary-driven wicking structures, and hybrid systems coupling photothermal or electrohydrodynamic effects. The following section summarizes and compares these representative strategies to provide a broader context for positioning our method.

Recent advances in thermal management have followed four main routes. i) surface modification to enhance evaporation: Berce et al. investigated laser-textured copper with tuned wettability to manage nanoparticle deposition and sustain boiling performance on structured heaters³⁰. Orman et al. investigated laser-generated grooves and microfins on copper, showing how geometry and roughness tailor nucleation and film dynamics in pool boiling³¹. Orman et al. further investigated laser process parameters (pulse duration/scan speed) to engineer multi-scale topographies that improve pool-boiling heat transfer³². ii) nanostructured coatings to improve heat transfer: Zhao et al. investigated ceramic-coated carbon nanotube (CNT) microstructures (via atomic layer deposition (ALD)) to create mechanically stable micro/nanoporous boiling interfaces with rapid liquid imbibition³³. Sen et al.

investigated Cu–TiO₂ nanoparticle coatings prepared by a hybrid method on copper to alter porosity and surface energy for enhanced pool boiling³⁴. Kumar et al. investigated two-step electrodeposited Cu–TiO₂ nanocomposite coatings to study how deposition strategy controls coating integrity and boiling behavior³⁵. iii) capillary-wicking liquid-transport structures: Chun et al. investigated hierarchical copper nanowire arrays with interconnected V-grooves to accelerate thin-film wicking for electronics cooling³⁶. Luo et al. investigated biomimetic “copper-forest” wicks for ultrathin heat pipes/vapor chambers to strengthen capillary lift in compact spreaders³⁷. Luo et al. also investigated copper-mesh wicks modified with a copper-forest structure to balance permeability and capillary pressure³⁸. iv) hybrid systems coupling photothermal or electrohydrodynamic effects: Wang et al. investigated graphene-assisted ionic-wind cooling for LEDs to intensify corona discharge and airflow over hot components³⁹. Xu et al. investigated a tri-needle/ring ionic-wind device embedded in LED bulbs to enhance forced convection around filaments⁴⁰. Cheng et al. investigated electrohydrodynamic gas pumps with aligned/offset electrodes for channel flows in electronics cooling⁴¹.

In comparison, these four routes largely i) operate in the contact/heated zone by altering nucleation and surface energy (surface modification, nanocoatings), ii) rely on sealed two-phase loops with designated wicks and working liquids to spread and return liquid (capillary structures), or iii) require external energy or fields to drive flow and heat removal (photothermal/EHD). By contrast, our square-cusp microstructure routes a precursor film from a non-contact reservoir so the droplet body remains a stable source while a thin film is continuously and directionally delivered to the hotspot, enabling sustained supply and programmable, anisotropic evaporative cooling without pumps, fans, light absorbers, or high voltage. This non-contact, lithography-defined routing avoids modifying the load-bearing surface, eases integration near chip hotspots, reduces fouling risk compared with in-zone texturing/coatings, and circumvents dry-out/choking limits typical of sealed wicks—thereby offering a distinct, energy-free pathway suited to intelligent, layout-aware heat management at device scale. At present time, our proposed microstructures just give a conceptual heat management scheme for

chips through liquid evaporation, which is simple and efficient. In future, we will do more work to put it into practical use.

We sincerely thank the respected reviewer once again for this constructive suggestion. In the revised manuscript (**Section 1 in Page 2, Fig. 1g in Page 3, Section 2.4 in Page 14 and Section 2.5 in Page 16**) and the Supporting information (**Supplementary Note 4 and Fig. S35**), we have provided a more thorough comparison with the existing literature, highlighted the specific limitations of prior work, and presented quantitative evidence of the advantages of our bulk-cusp design. All the corresponding modifications have been clearly marked in red for ease of checking.

2. The proposed mechanism of fluid spreading relies on simplified force balance equations. A more rigorous model, potentially incorporating computational fluid dynamics (CFD) simulations, would be beneficial to validate the proposed mechanism and provide deeper insights into the underlying physics.

Response 2: We sincerely thank the reviewer for this constructive suggestion. We fully agree that relying only on simplified force balance equations may limit the mechanistic depth of the analysis. A more rigorous model, especially based on computational fluid dynamics (CFD), is indeed valuable to validate the proposed mechanism and to provide deeper insight into the underlying fluid–structure interactions.

To this end, we performed CFD simulations using ANSYS Fluent 2022 R1, adopting the Volume of Fluid (VOF) method to resolve the multiphase dynamics at the liquid–air–solid interface. The simulation domain faithfully reproduced the geometric details of both the cross-cusp and square-cusp surfaces. Boundary conditions were set to match the experimental conditions: the static contact angle was fixed at the experimentally measured value, the surface tension coefficient of water was set as 0.072 N/m, and no-slip conditions were imposed at the solid boundaries. A mesh independence study was carried out to ensure that the results were insensitive to grid refinement.

Representative results are presented in **Figure R43** and **Figure R44**, which illustrate the time-resolved spreading process. Both cross-cusp and square-cusp surfaces exhibit directional spreading that follows the predesigned pathways, consistent with the experimental observations. Notably, the cross-cusp structure drives more pronounced anisotropic spreading with elongated advancement along preferred directions, whereas the square-cusp structure shows relatively more uniform, multi-directional expansion with faster spreading rates. The simulations also reveal transient curvature changes at the advancing meniscus near cusp tips, further supporting the geometric control of wetting evolution.

Figure R43. CFD simulation of time-resolved droplet spreading on cross-cusp microstructures.

Figure R44. CFD simulation of time-resolved droplet spreading on square-cusp microstructures.

Beyond reproducing the experimental phenomena, the simulations further clarify the intrinsic mechanism governing multi-directional spreading. The specially arranged cusp geometry induces asymmetric capillary forces at adjacent tips, which guide liquid motion along predetermined pathways. The resulting directional capillary traction overcomes local viscous resistance, leading to anisotropic advancement of the liquid front. In addition, the results demonstrate that the cusp configuration effectively regulates the transition between localized wetting and extended film spreading. The square-cusp surface promotes more uniform multi-directional flow, while the cross-cusp design strengthens directional confinement and enables preferential propagation along designed axes. These trends are consistent with the experimental observations, confirming that the programmable spreading behavior originates from the coordinated

effects of cusp-induced capillary forces, structural confinement, and geometric connectivity.

Overall, these CFD results validate our analytical framework by confirming that the observed spreading pathways emerge from the combined effects of capillary forces and structural confinement. In addition, the simulations enrich the physical understanding by providing direct visualization of interface deformation and spreading dynamics that complement the simplified model.

All corresponding results and explanations have been incorporated into the revised manuscript (**Section 2.3 in Page 11 and Fig. 3h in Page 12**) and Supporting Information (**Supplementary Figs. S31–S32 and Movies 6–7**), with new content highlighted in red for clarity. We sincerely thank the reviewer again for this valuable suggestion, which has significantly improved the rigor and completeness of our study.

3. The experimental section needs more detail regarding the setup, error analysis, and statistical significance of the results. The authors should provide more comprehensive characterization data for the fabricated microstructures, including information on feature size uniformity and defect density. Furthermore, the performance gains in lubrication and thermal management must be benchmarked against state-of-the-art technologies to quantify the improvement achieved by the proposed design.

Response 3: We sincerely thank the reviewer for this insightful and constructive comment. The request for more detailed descriptions of the experimental setup, error analysis, and statistical significance, along with comprehensive characterization of the fabricated microstructures and benchmarking against state-of-the-art technologies, is highly valuable. These aspects are indeed essential for ensuring the rigor, transparency, and credibility of the reported findings, as they not only strengthen the reliability of the experimental data but also place the demonstrated performance in a broader technological context.

- **Experimental Details and Error Analysis**

1) Materials

Ethanol–deionized water mixtures with various volume concentrations (0%, 20%, 40%, 60%, 80%, and 100%) were prepared for the experiments. Absolute ethanol (analytical grade) was purchased from Shanghai Macklin Biochemical Co., Ltd. (Shanghai, China), and deionized water was supplied by Shanghai Aladdin Biochemical Technology Co., Ltd. (Shanghai, China). In addition, three oils with different viscosities were selected for comparison, including white oil, paraffin oil, and castor oil, all of which were obtained from Shanghai Aladdin Biochemical Technology Co., Ltd. All reagents were of analytical grade and were used without further purification.

2) Fabrication of Bulk-Cusp Micro-structured Surfaces

We have enriched the description of the fabrication of bulk-cusp micro-structured surfaces by providing additional details of the photolithographic process. Specifically, we now state that silicon wafers were first immersed in a freshly prepared piranha solution ($\text{H}_2\text{SO}_4:\text{H}_2\text{O}_2 = 7:3$) for 30 min, followed by sequential rinsing with deionized water, ethanol, and acetone, and subsequently baked at 100 °C for 10 min to remove residual contaminants. A positive photoresist (AZ4620, MicroChemicals, Germany) was then spin-coated at 3000 rpm using a Laurell WS-650-23 coater, and prebaked in two steps (5 min at 60 °C and 20 min at 90 °C) to improve adhesion and uniformity. UV exposure was carried out using a SUSS MA6 mask aligner, and the patterned resist was developed in AZ-400K solution. Deep reactive ion etching was performed to obtain the bulk-cusp geometry, after which the residual resist was completely removed by rinsing with acetone, ethanol, and deionized water, yielding sharply defined cusp features.

In addition, we have added an error and reproducibility analysis focusing on the contact angle measurements, which are central to evaluating the wettability of the fabricated surfaces. For each surface type, at least three droplets were tested at independent positions on the wafer, and each droplet was measured three times to reduce random error. These results not only demonstrate the reliability of the surface treatment but also provide quantitative support for the reproducibility and statistical

validity of the experimental observations.

3) Multi-directional Liquid Spreading Experiments

We have expanded this section by incorporating additional liquid systems beyond deionized water, including ethanol–water mixtures with concentrations of 20%, 40%, 60%, 80%, and 100% (v/v) to examine the influence of surface tension reduction, as well as lubricating oils with different viscosities to assess the effect of viscosity on spreading dynamics. These liquids were dispensed in the same volume of 5 μL using calibrated micropipettes to ensure consistency. Moreover, to better reflect practical liquid transport scenarios, we introduced a continuous injection condition, where a microinjection pump (ZS100-01, Baoding Chuangrui Precision Pump Co., Ltd., China) delivered deionized water at a constant rate of 10 $\mu\text{L}/\text{min}$. This allowed us to compare instantaneous spreading after single-drop deposition with steady-state dynamics under sustained inflow.

In addition, we implemented systematic error and reproducibility analyses. For each condition, each droplet experiment was repeated three times. The recorded spreading velocities and normalized spreading lengths were averaged, with error bars representing one standard deviation. The measurement uncertainty in droplet volume was controlled within $\pm 0.1 \mu\text{L}$, while the frame rate calibration of the high-speed camera introduced an estimated temporal error below 2 ms. Statistical analysis confirmed that the differences between different liquids, are significant, thereby supporting the robustness of our conclusions. These additions provide both broader experimental validation across liquid types and rigorous statistical support for the reproducibility of the spreading measurements.

4) Tribological Experiments

We have expanded this section by adding further details on pre-test preparation and extended testing conditions. Before each experiment, the contact surfaces of both the copper pin and the steel disk were carefully polished and cleaned to remove residual contaminants, after which the silicon wafer samples were embedded into the grooves and fixed with PDMS to ensure surface flushness. In addition to the bare surface, a

cross-patterned surface without cusp tips was tested as a control to further clarify the contribution of cusp features. The operating conditions were enriched to include three sets of scenarios: (i) a baseline test at 30 N load and 4 Hz frequency for 10 minutes; (ii) load variation at a fixed frequency of 4 Hz with normal loads of 10, 20, 30, 40, and 50 N; and (iii) frequency variation at a fixed load of 30 N with sliding frequencies of 2, 3, 4, 5, and 6 Hz. In all cases, ~50 μL of deionized water was applied to the contact zone prior to testing, and the real-time coefficient of friction was continuously recorded.

Furthermore, we added more rigorous error control and statistical assessment to validate the results. Each condition was tested with at least three independent repetitions, and the data were summarized as mean values with standard deviations. Statistical analysis confirmed that the observed differences in friction coefficients among the bare, cross, and cross-cusp surfaces are meaningful and reproducible. These additions strengthen the robustness and reliability of the tribological findings.

5) Thermal Management Experiments

We have expanded this section by adding further details on the thermal management experiments. The heating platform was a microcomputer-controlled unit (JF-956B, JFTOOIS, China) maintained at 50 $^{\circ}\text{C}$ to ensure accurate and stable thermal conditions. Prior to each test, the silicon wafer samples were thoroughly cleaned with ethanol and deionized water, followed by nitrogen drying, to remove any residual dust or contaminants. In addition to the bare flat wafer, a square-patterned surface without cusp tips was also included as a control, allowing us to further evaluate the specific contribution of cusp features in thermal performance. For the evaporation experiments, ~5 μL droplets of deionized water were carefully deposited using a micropipette, with both single-drop and repeated deposition (every 30 s) scenarios tested. Throughout the tests, infrared thermography continuously monitored the temperature evolution, and the data were processed with custom MATLAB scripts to extract spatially averaged surface temperatures.

Furthermore, we added more rigorous error control and statistical assessment. Each experimental condition was repeated at least three times using independently

prepared samples. The results are presented as mean values with standard deviations, and statistical analysis confirmed that the observed differences in cooling performance between bare, square, and square-cusp surfaces are significant and reproducible. These additions enhance the robustness and reliability of the thermal management findings.

6) Contact angle, surface tension, and viscosity measurements

Contact angle measurements were performed using a contact angle goniometer (OCA20, Dataphysics Co., Germany). Surface tension of the liquids was measured with a tensiometer (DSA100S, KRUSS, Germany), while dynamic viscosity was determined using a rotational viscometer (DV2T, Brookfield, USA). For each liquid or surface type, at least three independent measurements were carried out. Specifically, contact angle measurements were performed on three droplets placed at separate positions on the wafer, and each droplet was measured three times to minimize random error. Similar procedures were followed for both surface tension and viscosity tests to ensure consistency.

7) Morphology Characterization

We have expanded this section by including additional details on error control and statistical assessment. For SEM imaging, multiple regions across each wafer were examined to verify the uniformity of microstructure morphology, and representative images were selected from consistent observations. For laser confocal microscopy, at least five independent measurement areas were analyzed for each sample to quantify feature height and wear profile depth. The results were summarized as mean values with standard deviations to account for sample-to-sample variations. To further ensure reproducibility, morphology characterization was performed on wafers prepared in different fabrication batches, and consistent results confirmed that the microstructures are fabricated with high fidelity and stability. Statistical analysis confirmed that variations in feature size and wear profile depth remained within narrow limits, indicating reliable measurement accuracy and meaningful differences between the tested surfaces.

● Characterization of Feature Size Uniformity and Defect Density

The characteristic dimensions of the fabricated microstructures are of primary importance, as they govern wetting dynamics and strongly influence subsequent tribological and thermal behaviors. To ensure accuracy, we performed quantitative measurements on 500 microstructures for each category, and the results are summarized in **Figure R45**. The designed cusp spacing, cross length, and square length were 3 μm , 20 μm , and 27 μm , respectively. The actual measured averages were $2.98 \pm 0.0052 \mu\text{m}$, $19.93 \pm 0.0153 \mu\text{m}$, and $26.94 \pm 0.0187 \mu\text{m}$, demonstrating excellent agreement with the design. ImageJ software was employed to extract values from high-resolution microscopy images, and the large dataset was analyzed statistically. Gaussian fitting of the histograms revealed narrow distribution widths, with the peak values closely centered on the design targets. Scatter plots further confirmed that the measurement points clustered tightly around the mean, highlighting the high reproducibility and dimensional precision achieved by the photolithographic process.

Figure R45. Statistical analysis of feature size uniformity of the fabricated microstructures. a Histogram and Gaussian fitting of cusp spacing. **b** Histogram and Gaussian fitting of cross length. **c** Histogram and Gaussian fitting of square length. **d** Scatter plots of the three feature sizes.

In addition to dimensional uniformity, defect density was systematically assessed, as illustrated in **Figure R46**. For both cross-type and square-type microstructures, five independent regions were analyzed, each including approximately 500 microstructures. The defects identified mainly consisted of missing units, incomplete etching, and edge irregularities. Statistical analysis revealed that the average defect density for both types of structures consistently remained in the narrow range of 0.4–0.6% across different regions. The low frequency of defects confirms that the microfabrication process is robust and that occasional imperfections do not significantly compromise the overall structural integrity.

Figure R46. Evaluation of defect density in the fabricated microstructures. a Defect density distribution of cross-cusp microstructures. **b** Defect density distribution of square-type microstructures.

Taken together, these results provide a comprehensive validation that the fabricated microstructures not only reproduce the intended design dimensions with high fidelity but also maintain extremely low defect densities. The Gaussian-distributed size data and consistent defect statistics highlight the reliability of the lithography-based fabrication process, ensuring the structural stability and reproducibility necessary for subsequent functional investigations.

- **Evaluation of Lubrication and Thermal Management Relative to State-of-the-Art**

The respected reviewer’s suggestion to benchmark our results in lubrication and thermal management against advanced strategies is constructive and meaningful, as it

helps situate our work within the broader context of tribology and thermal regulation research. Addressing this point allows us not only to highlight the quantitative merits of our design but also to clarify its distinct advantages compared with conventional approaches.

To address this issue concisely here, our bulk-cusp design for lubrication achieves a stable friction coefficient of ~ 0.2 compared to ~ 0.3 on bare surfaces, corresponding to a $\sim 35\%$ reduction—exceeding the $<30\%$ benchmarks typically reported for coatings, additives, texturing, or strengthening. This advantage arises from lubricant replenishment via non-contact capillary pathways, ensuring both energy-free operation and long-term stability. In thermal management, the square-cusp structure enables directional spreading of a precursor film while keeping the droplet body stationary, thereby providing a continuous liquid supply for efficient evaporative cooling. Importantly, there are currently no directly comparable studies in the literature that offer quantitative benchmarks equivalent to our results; rather, the novelty of our approach lies in its mechanism of programmable, directional, and sustained precursor-film supply. Prior advances have mainly focused on surface modification to enhance evaporation, nanostructured coatings to improve heat transfer, capillary-driven wicking structures, or hybrid systems coupling photothermal or electrohydrodynamic effects. Thus, our square-cusp mechanism represents a conceptual innovation with strong potential for intelligent chip-level thermal regulation. A detailed literature review and quantitative benchmarking have already been provided by **Reviewer #3 in Major Concerns 1** (Page 47 in *Response to Comments*) for comprehensive comparison with state-of-the-art technologies.

We sincerely thank the respected reviewer again for these constructive comments, which have helped us improve the clarity and completeness of our work. In the revised manuscript (**Section 1 in Page 2, Section 2.1 in Page 5, Section 2.4 in Page 14, Section 2.5 in Page 16, Section 4 in Pages 17–20, and Fig. 1g in Page 3**) and the Supporting information (**Supplementary Note 4, Figs. S6–S7, and Fig. S35**), we have provided additional details on the experimental setup, error analysis, and statistical

treatment of the results, as well as comprehensive data on feature size uniformity and defect density of the fabricated microstructures. Furthermore, we have benchmarked the lubrication and thermal management performance against representative state-of-the-art methods to better highlight the advantages of our approach. All corresponding revisions have been clearly marked in red in the revised files.

4. The manuscript is generally well-written, but some sections could be clarified. The distinction between cross-cusp and square-cusp structures, as well as their roles, should be emphasized.

Response 4: We sincerely thank the respected reviewer for this constructive comment. The request for a clearer mechanistic explanation and for distinguishing the respective functions of the cross-cusp and square-cusp microstructures is very valuable and has guided us to refine and strengthen our discussion.

Mechanism analysis shows that capillary forces generated between adjacent cusps trigger the separation and guided extension of the precursor film, while the coverage ratio of precursor-film-accessible area determines whether effective traction can be able to drag the droplet body. Quantitative image analysis reveals that the cross-cusp microstructure possesses a relatively high open-area ratio (75%), compared to 45% in the square-cusp counterpart. The larger open area ensures continuous connectivity of precursor films, facilitating their extension and forming wettable pathways that efficiently drag the droplet body along the designed axes. Consequently, the droplet body on cross-cusp surfaces exhibits rapid, anisotropic spreading with high directional fidelity. In contrast, the smaller open-area ratio of square-cusp surfaces limits precursor film connectivity, leading to fragmented film coverage and weaker traction on the droplet body. As a result, droplet body spreading is strongly constrained, while precursor film remains under structural guidance.

These mechanistic differences also explain the distinct functional advantages of the two designs. On cross-cusp surfaces, the ability to generate strong traction and accelerate droplet body spreading directly enhances lubrication by ensuring continuous

delivery of liquid to the contact region, thereby reducing friction and suppressing wear in tribological systems. On square-cusp surfaces, although the traction on the droplet body is weaker, the regulation of the precursor film leads to uniform and repeatable film coverage. This promotes stable evaporation cooling and sustained heat dissipation, making it highly relevant for thermal management applications.

Taken together, the cross-cusp and square-cusp surfaces highlight two complementary functions: one facilitates efficient liquid delivery, while the other stabilizes precursor-film spreading. These insights provide a mechanistic foundation for rationally designing microstructured surfaces to target specific application scenarios.

Once again, we sincerely thank the reviewer for raising this important point. In the revised manuscript (**Section 2.3 in Page 11**), we have clarified the mechanistic distinctions, emphasized the different application-related implications of the two structures, and highlighted all revisions in red for the reviewer's convenience.

5. Figures should be improved with labels and self-explanatory captions. It would be beneficial if authors present the exact figure in the main text, along with data and schematic illustrations from the same structures used (with the same orientation and spreading mode). For example, Figures 3a, d, and e have the same orientation of cusp, but Figure 3b has a different one.

Response 5: We sincerely thank the reviewer for this valuable suggestion. We fully agree that figures with well-defined labels and self-explanatory captions are critical for effective scientific communication. This recommendation also echoes a related concern raised by **Reviewer #2**, further highlighting the importance of figure clarity and consistency throughout the manuscript and Supporting Information.

After carefully reviewing all figures, we identified that certain schematics lacked sufficient annotation or consistent orientation. We have accordingly revised the figures to enhance their clarity. Specifically, we added labels, directional indicators, and more descriptive captions to multiple figures to ensure they can be more easily interpreted without referring extensively to the main text.

We particularly appreciate the reviewer's detailed observation that Figs. 3a, 3d, and 3e share the same cusp orientation, while Fig. 3b differs. This inconsistency was indeed due to our oversight. To address this, we have updated the orientation of **Fig. 3b**, and the revised version is now presented as **Figure R47**, replacing the original version in the manuscript to ensure visual consistency across all illustrations.

Figure R47. Schematic illustrating precursor film spreading along the sidewalls and bottom grooves due to asymmetric capillary forces on the C-mode II surface.

Additionally, due to the diversity of spreading modes and geometric configurations investigated in this work, we were unable to include all data and diagrams in the main text. Several figures were placed in the Supporting Information to supplement the primary discussion. We have now re-examined these figures as well—such as those showing droplet spreading morphologies and interface dynamics—and revised those with unclear layouts or insufficient explanatory details by adding appropriate labels and improving caption content.

All relevant revisions in both the manuscript (**Fig. 3b**) and Supporting Information (**Supplementary Figs. S10–S11 and Figs. S14–S15**) have been clearly marked in red font for the reviewer's convenience. Once again, we thank the reviewer for this insightful recommendation, which has helped improve the overall presentation quality of our work.

6. The method for quantifying the spreading length ('L') requires significant clarification. As defined, 'L' represents the spreading length along the x^+ , x^- , y^+ , or y^- axes. However, this definition is unclear in cases where spreading is not

strictly unidirectional, such as the quasi-circular spreading observed with cross-cusp structures and the square-like spreading with square-cusp structures. The manuscript lacks a precise explanation, with illustrative examples, of how 'L' is determined for each spreading mode (C-pinning, C-mode I-IV, S-pinning, S-mode I-IV).

Response 6: We thank the respected reviewer for this valuable comment regarding the definition of spreading length L . This is particularly relevant in cases with quasi-circular or square-like spreading. We agree that the original description lacked clarity in these contexts. In response, we have revised the manuscript to clarify the calculation approach and added schematic illustrations to aid understanding and reproducibility.

In response to the respected reviewer's observation regarding quasi-circular and square-like spreading patterns on cross-cusp and square-cusp surfaces, we would like to clarify that the spreading length 'L' in our study refers to the droplet body rather than the precursor film. This choice reflects both the intended macroscopic guidance function of the designed structures and the practical feasibility of measurement. The droplet body forms a stable and well-defined contour, enabling consistent quantification, while the precursor film spreads rapidly and diffusely, making it unsuitable for accurate assessment.

To quantify directional spreading, we defined the spreading length L (L_{x^+} , L_{x^-} , L_{y^+} , and L_{y^-}) as the maximum projected distance from the droplet's geometric center (releasing point) to its contour along the four principal directions (x^+ , x^- , y^+ , and y^-). We acknowledge that the previous description was imprecise—it is not the actual spreading length along these axes, but the projection of the contour onto these directions, as clarified in the revised manuscript and illustrated in **Figure R48**. The geometric center was determined at the initial moment of contact, when the droplet exhibits a near-circular shape, providing the location reference for measurement. To enable consistent comparisons across different droplet volumes and substrate structures, each directional length was normalized by the initial droplet radius R , measured at the initial contact moment. These 4 dimensionless parameters collectively reflect the relative spreading

extent and anisotropic wetting behavior of the droplet body on the surface with microstructures:

$$\tilde{L}_{x+} = \frac{L_{x+}}{R}, \tilde{L}_{x-} = \frac{L_{x-}}{R}, \tilde{L}_{y+} = \frac{L_{y+}}{R}, \tilde{L}_{y-} = \frac{L_{y-}}{R} \quad (\text{R9})$$

Figure R48. Schematic illustration of the definition of spreading lengths along the 4 principal directions (x+, x-, y+, and y-).

To clarify the quantification method for spreading length L , **Figure R49** illustrates how L is measured on cross-cusp and square-cusp surfaces under various spreading modes. As the reviewer rightly noted, the original text may have caused ambiguity; the values of L are not actual axial lengths but projected distances from the droplet's geometric center to the outermost boundary along the four principal directions (x+, x-, y+, and y-).

Figure R49. Schematic demonstration of spreading length quantification on cross-cusp and square-cusp surfaces under various spreading modes.

While this method facilitates consistent quantification, it is limited by its coarse

resolution and inability to reflect full contour information. To address this, we introduce two additional indicators in the revised manuscript. These improvements, along with the updated explanation and schematic, are highlighted in red in both the main text (**Section 2.2 in Pages 5–6**) and Supporting Information (**Supplementary Figs. S9–S12**).

The authors are advised to:

1. Provide an unambiguous definition of how 'L' is measured for each spreading mode, including a diagram illustrating the measurement process for each distinct case. Additionally, which droplet radius is considered? Of a spread droplet or one calculated based on the droplet volume.

Response 1: We appreciate the respected reviewer's request for a clearer and more unambiguous definition of how the spreading length L is determined. As elaborated in our response by **Reviewer #3 in Major Concerns 6** (Page 69 in *Response to Comments*), L is defined as the projected distance from the geometric center of the droplet body to the outermost boundary along each of the four principal directions ($x+$, $x-$, $y+$ and $y-$). This definition does not refer to the full length along the axes but to the maximum extent of the droplet contour projected onto each axis, as clarified and illustrated in **Figure R48**. We acknowledge that this distinction was insufficiently stated in the original manuscript.

To improve clarity, **Figure R49** shows representative measurement schematics of L for different spreading modes on cross-cusp and square-cusp surfaces. These examples illustrate how the axis-based method is applied across varying droplet shapes. To eliminate the influence of initial volume and enable standardized comparison across surfaces, all directional lengths (L_{x+} , L_{x-} , L_{y+} , and L_{y-}) are normalized by the initial droplet radius R , defined as the radius of the droplet body at the moment of contact with the substrate. This radius is extracted from the top-view projection of the droplet, which approximates a circle prior to spreading, and thus provides a robust reference length scale. In all experiments, the dispensed droplet volume was kept constant at

approximately 5 μL . By assuming the droplet to be an ideal sphere before contact, the calculated radius based on volume agrees closely with the observed radius from top-view images, with discrepancies falling within the experimental error range.

Figure R50 and **Figure R51** present the time-dependent evolution of normalized spreading lengths (L/R) along four principal directions for each mode on the cross-cusp and square-cusp surfaces, respectively. These plots allow direct visualization of directional spreading kinetics and facilitate comparative analysis of anisotropy and dynamic behavior across different microstructure types.

All relevant descriptions and schematics have been updated in the revised manuscript (**Section 2.2 in Pages 5–6**) and Supporting Information (**Supplementary Figs. S9–S12**), with new content highlighted in red for the reviewer’s convenience.

Figure R50. Time-dependent spreading of the droplet body on cross-cusp microstructure under various spreading modes.

Figure R51. Time-dependent spreading of the droplet body on square-cusp microstructure under various spreading modes.

2. Consider alternative measurement approaches that are less sensitive to the shape of the spreading pattern. For instance, an effective radius (based on the spread area) or a direct area measurement might be more appropriate for non-unidirectional spreading (e.g., $A_{\text{drop}}/A_{\text{spread}}$ where A_{drop} is the base area of droplet and A_{spread} is the spreading area).

Response 2: We sincerely thank the reviewer for this valuable suggestion. As noted, the current axis-based method for quantifying spreading length L is more appropriate for unidirectional spreading, but may not fully capture the complexity of more symmetric or irregular spreading patterns, where axis-dependent measurements could introduce shape-related biases. The reviewer’s proposed alternatives—such as effective radius or area-based ratios ($A_{\text{drop}}/A_{\text{spread}}$)—are indeed more suitable for non-unidirectional spreading, as they are less sensitive to geometric anisotropy, and provide a complementary perspective to our current analysis.

To address this issue, we propose two complementary strategies aimed at improving the fidelity and completeness of spreading quantification:

(1) Enhanced resolution via eight-directional normalized length analysis

We extend the original axis-based method by incorporating four additional diagonal directions (45° , 135° , 225° , and 315°) alongside the standard Cartesian axes (x^+ , x^- , y^+ and y^-). For each direction θ , the spreading length L_θ is defined as the linear distance from the droplet's geometric center to its outermost boundary along that direction. To eliminate volume-related variability, spreading lengths are normalized by the initial droplet radius R , measured at the moment of contact when the droplet approximates a circular cap:

$$\tilde{L}_\theta = \frac{L_\theta}{R}, \theta \in \{x^+, x^-, y^+, y^-, 45^\circ, 135^\circ, 225^\circ, 315^\circ\} \quad (\text{R10})$$

The spreading lengths along the four new diagonal directions are denoted as L_1 , L_2 , L_3 , and L_4 , corresponding to 45° , 135° , 225° , and 315° , respectively. The schematic of this measurement approach is provided in **Figure R52**, which visually demonstrates how directional lengths are extracted in both cardinal and diagonal directions. This figure serves to clarify the extended quantification method and ensure reproducibility across various spreading geometries.

Figure R52. Schematic illustration of the definition of spreading lengths along the eight directions (x^+ , x^- , y^+ , y^- , 45° , 135° , 225° , and 315°).

To better visualize how the eight-directional method is applied in practice, we constructed a set of schematic illustrations. As shown in **Figure R53**, this method captures the droplet's spreading front along both the Cartesian and diagonal axes, enabling improved angular resolution in quantification. Each panel depicts the geometric center of the droplet and the measured spreading lengths in eight directions, labeled as L_{x^+} , L_{x^-} , L_{y^+} , L_{y^-} , L_1 , L_2 , L_3 , and L_4 .

Figure R53. Schematic demonstration of spreading length quantification on cross-cusp and square-cusp surfaces under various spreading modes.

To facilitate comparative analysis, we further applied this eight-directional quantification method to various spreading modes on both the cross-cusp and square-cusp surfaces. As shown in **Figure R54** and **Figure R55**, the normalized spreading lengths along all eight directions (x^+ , x^- , y^+ , y^- , 45° , 135° , 225° , and 315°) are plotted for each mode, enabling a more comprehensive assessment of anisotropy and directional preference across microstructures.

Figure R54. Time-dependent normalized spreading lengths (L/R) of the droplet body along eight directions on cross-cusp microstructure under various spreading modes.

Figure R55. Time-dependent normalized spreading lengths (L/R) of the droplet body along eight directions on square-cusp microstructure under various spreading modes.

To further illustrate the advantages of this eight-directional quantification method, it compares the normalized spreading lengths of droplets along eight directions for both cross-cusp and square-cusp surfaces under various spreading modes in **Figure R56**. Each radar chart reveals the anisotropy and directional preferences in spreading behavior. For example, in mode I, the cross-cusp surface exhibits pronounced elongation along the $x+$ direction, while the square-cusp structure shows a more isotropic pattern.

These comparative visualizations not only highlight the distinct multi-directional spreading characteristics of the two surface types but also underscore the effectiveness of this method in capturing both the magnitude and anisotropy of spreading. This enables a comprehensive and intuitive assessment of spreading behavior across diverse modes and structures.

Figure R56. Radar plots comparing normalized spreading lengths (L/R) along eight directions for cross-cusp and square-cusp microstructures under various spreading modes.

(2) Quadrant-based area normalization and equivalent radius calculation

To further evaluate global spreading morphology, we introduce a quadrant-based area quantification strategy as the respected reviewer suggested. At initial contact, the droplet is approximated as a circle with radius R , and the substrate is divided into four quadrants using a right-angle coordinate system centered at the droplet's geometric center. Each quadrant initially holds area $S = \frac{1}{4} \pi R^2$.

After spreading, the footprint is segmented into quadrant areas S_1, S_2, S_3, S_4 , which are normalized relative to S :

$$\tilde{S}_i = \frac{S_i}{S}, \quad i \in \{1, 2, 3, 4\} \quad (\text{R11})$$

The total spreading area is defined as:

$$A_{\text{spread}} = S_1 + S_2 + S_3 + S_4 \quad (\text{R12})$$

In addition, the total area can be used to define an equivalent spreading radius R' , assuming the spread footprint is approximated by a circle:

$$R' = \sqrt{\frac{A_{\text{spread}}}{\pi}} \quad (\text{R13})$$

To better illustrate the quadrant-based area quantification strategy, a schematic diagram is presented in **Figure R57**. Upon spreading, the footprint is divided into four quadrants, each corresponding to an area S_1, S_2, S_3, S_4 , which are normalized by the initial area defined by radius R . This visualization outlines how the total normalized spreading area and the equivalent spreading radius R' are derived, enabling a robust and geometry-tolerant quantification of non-axisymmetric spreading behavior.

Figure R57. Schematic illustration of the quadrant-based area normalization method used to calculate directional spreading areas and the equivalent radius R' .

To visualize the practical implementation of the quadrant-based area normalization method, representative steady-state footprints were extracted for various structure–mode combinations. As shown in **Figure R58**, each panel displays the final spreading pattern segmented into four quadrant areas (S_1 – S_4), centered at the droplet’s geometric center. This framework allows direct mapping of the spatial distribution of spreading into quantitative areal metrics, thereby facilitating standardized, geometry-independent comparisons across diverse structural designs and spreading modes.

Figure R58. Schematic illustrations of the quadrant-based area normalization method applied to different spreading modes on cross-cusp and square-cusp surfaces.

To further demonstrate the applicability of the quadrant-based area method, **Figure R59** and **Figure R60** present the time-resolved normalized spreading areas across four quadrants for the cross-cusp and square-cusp microstructures, respectively. The cross-cusp microstructure exhibits overall larger normalized spreading areas in all quadrants, reflecting a more effective guidance of droplet body spreading. In contrast, the square-cusp microstructure shows relatively limited spreading extent, with slower growth and reduced anisotropy. These results highlight the superior directional spreading performance enabled by the cross-cusp design.

Figure R59. Time-dependent normalized spreading areas (S_i/S) in four quadrants on cross-cusp microstructure under various spreading modes.

Figure R60. Time-dependent normalized spreading areas (S_i/S) in four quadrants on square-cusp microstructure under various spreading modes.

To quantitatively evaluate the global spreading capacity under various modes and structures, it presents the calculated equivalent spreading radius R' using the proposed normalization method in **Figure R61**. It is evident that the cross-cusp microstructure consistently exhibits a larger R' than the square-cusp microstructure, indicating more extensive expansion in all quadrants.

Figure R61. Equivalent spreading radius R' under different spreading modes on cross-cusp and square-cusp microstructures.

In summary, these two methods are designed to complement each other: the first provides high-resolution angular information, while the second offers a robust global metric of spreading magnitude and symmetry. Together, they significantly improve the

reliability and interpretability of spreading measurements, particularly for irregular modes.

We thank the reviewer once again for raising this important point. The combination of these two strategies—directional length extraction and quadrant-based area analysis—offers a robust framework for capturing both localized and global characteristics of complex droplet spreading. All corresponding additions and revisions have been clearly marked in the revised manuscript (**Section 2.2 in Pages 5–8, Figs. 2h–j in Page 7**) and Supporting Information (**Supplementary Note 2, Figs. S9–S15**) using red text.

3. Provide an unambiguous definition of how 'L' is measured for each spreading mode, including a diagram illustrating the measurement process for each distinct case.

Response 3: We appreciate the respected reviewer's request for a clearer and more rigorous definition of the spreading length L , which is essential for reliable characterization of anisotropic wetting behavior. As detailed in **Major Concerns 6 provided by Reviewer #3** (Page 69 in *Response to Comments*) and **Advised Concerns 1 provided by Reviewer #3** (Page 72 in *Response to Comments*), we have clarified in the revised manuscript that L is defined as the projected distance from the geometric center of the droplet body to its outermost boundary along the four principal directions: x^+ , x^- , y^+ and y^- .

To aid reproducibility, we have added **Figure R48**, which schematically illustrates how the geometric center is located and how L is measured along each axis. Furthermore, **Figure R49** presents representative examples of this measurement applied to various spreading modes on both cross-cusp and square-cusp surfaces. These examples correspond to the full range of spreading behaviors discussed in the manuscript. To provide quantitative context, we also include **Figure R50** and **Figure R51**, which show the time-dependent evolution of normalized spreading lengths (L/R) for each mode, enabling comparison of dynamic spreading behaviors across structures.

This axis-based method offers practical advantages in terms of clarity and

consistency. However, as discussed in **Major Concerns 6 provided by Reviewer #3** (Page 69 in *Response to Comments*), we also recognize its limitations in capturing angular or curved spreading contours. To address this, we have proposed two complementary evaluation strategies in the revised manuscript to improve angular resolution and spatial integration.

All related revisions and schematic illustrations are highlighted in red in both the main text (**Section 2.2 in Pages 5–8, Figs. 2h–j in Page 7**) and Supporting Information (**Supplementary Note 2, Figs. S9–S15**) to ensure full transparency.

4. Consider alternative measurement approaches that are less sensitive to the shape of the spreading pattern. For instance, an effective radius (based on the spread area) or a direct area measurement might be more appropriate for non-unidirectional spreading.

Response 4: We thank the reviewer for this valuable suggestion, which reiterates the point raised in **Advised Concerns 2 provided by Reviewer #3** (Page 74 in *Response to Comments*) regarding the need for quantification methods that are less sensitive to spreading shape.

Following this recommendation, we developed an alternative analysis framework termed quadrant-based area normalization and effective radius calculation, aimed at capturing complex non-unidirectional spreading behavior more comprehensively. As illustrated in **Figure R57**, the spreading area is divided into four quadrants to obtain spatially resolved normalized areas. Schematic application of this method to representative droplet profiles under different spreading modes is shown in **Figure R58**. Quantitative angular distributions of the quadrant-normalized areas are compared across two microstructures in **Figure R59** and **Figure R60**. To summarize the overall spreading extent, the effective radius derived from the total spread area is further computed, with comparative results presented in **Figure R61**. This method provides a robust, geometry-insensitive metric for evaluating spreading performance, and complements the directional-length-based approach introduced earlier.

In summary, these two alternative methods (*Enhanced angular resolution via eight-directional normalized length analysis* and *Quadrant-based area normalization and equivalent radius calculation*) jointly enhance our ability to characterize multi-directional and irregular spreading patterns with improved accuracy and robustness. While the first method improves angular detail in directional measurements, the second offers a spatially integrated metric that reduces dependence on local edge morphology.

All revisions and additional illustrations have been incorporated into the revised manuscript (**Section 2.2 in Pages 5–8, Figs. 2h–j in Page 7**) and Supporting Information (**Supplementary Note 2, Figs. S9–S15**), with changes clearly highlighted in red.

5. Justify the chosen measurement metric, explaining why it is the most relevant for characterizing the fluid spreading behaviour under investigation, particularly concerning the goal of directional control. If square spreading is observed, its functional significance should be articulated.

Response 5: We sincerely appreciate the reviewer’s thoughtful comment. The issues raised center on the rationale behind the chosen measurement metric, its relevance to directional control, and the functional significance of square-like spreading patterns. Below, we provide a point-by-point clarification:

(1) Justification of the chosen measurement metric (L_{x+} , L_{x-} , L_{y+} , and L_{y-}):

To characterize directional spreading, we adopted a length-based metric that measures the droplet’s extent along four principal Cartesian axes ($x+$, $x-$, $y+$ and $y-$), normalized by the initial droplet radius R . This metric offers several advantages:

- Directional sensitivity: The selected axes match the geometric symmetry of the surface microstructures, making directional trends easy to resolve.
- Measurement reliability: The droplet forms a stable and reproducible contact perimeter, facilitating consistent data acquisition.
- Functional relevance: The directional lengths reflect the extent of spreading guided by surface anisotropy, directly aligned with our study’s goal of

evaluating directional control.

- Literature precedent: Similar axis-based approaches are widely used in studies of anisotropic wetting and guided transport (see refs. ¹⁰⁻¹⁴).

However, we acknowledge that this metric may not capture non-axis-aligned deformation or irregular contours in certain spreading modes. To address this limitation and improve fidelity, we implemented two complementary methods:

- Enhanced angular resolution via eight-directional normalized length analysis: Incorporates four diagonal directions (45° , 135° , 225° , and 315°) to enhance angular resolution.
- Quadrant-based area normalization and effective radius calculation: As described in our response to **Advised Concerns 2 provided by Reviewer #3** (Page 74 in *Response to Comments*), this method divides the droplet footprint into four quadrants, and computes normalized spreading areas (S_1/S , S_2/S , etc.) and equivalent radius (R') based on total area. It provides a more integrative view of asymmetric spreading, particularly in irregular patterns.

Together, these measurements allow a more comprehensive and flexible analysis of droplet behavior across varied surface geometries and spreading modes.

(2) Relevance of this metric to the goal of directional control:

The axis-based projection method, which quantifies spreading lengths along four orthogonal directions (L_{x+} , L_{x-} , L_{y+} , and L_{y-}), is particularly well-suited for evaluating directional control in our system. Its relevance is reflected in three key aspects:

- Alignment with surface anisotropy: The cross-cusp and square-cusp microstructures are designed to guide liquid along specific directions in x and y axes. Measuring length differences along x and y axes directly reveals whether the geometry achieves directional preference.
- Sensitivity to anisotropic behavior: This metric effectively captures asymmetric spreading, allowing us to quantify how strongly a surface induces directional transport.
- Applicability to dynamic analysis: It supports both static and time-resolved

measurements, enabling insight into how rapidly and effectively directional control develops during spreading.

In summary, this method provides a direct, quantitative link between surface design and directional wetting behavior, making it well-suited to assess the core objective of our study.

(3) Functional interpretation of observed square spreading:

Our measurement strategy focuses on the droplet body, not the precursor film, due to both physical relevance and experimental feasibility. Mechanistically, the precursor film spreads rapidly along anisotropic microstructures upon impact, driven by capillary gradients and structural cues. This film, though transient and morphologically unstable, plays a guiding role by exerting traction on the main droplet, steering its directional spreading. However, its ill-defined and dynamic boundary makes it unsuitable for consistent quantification.

In contrast, the droplet body forms a stabilized, well-defined contour that faithfully reflects the integrated effect of structural guidance. Measuring its spreading behavior offers a robust and representative readout of the directionality imparted by surface design. Occasionally observed square-like spreading on square-cusp surfaces (e.g., S-mode III and IV in **Figure R62**) arises from symmetric pinning effects along the four orthogonal edges. Rather than being an artifact, such patterns indicate a reproducible, structure-induced outcome. Within our axis-based framework, this morphology:

- Enables objective, standardized comparison with other spreading modes;
- Reflects uniform multi-directional confinement, useful in applications requiring isotropic coverage or fluid localization.

Figure R62. Square-like spreading of the droplet body on square-cusp surfaces in (a) S-mode III and (b) S-mode IV.

Thus, square spreading is interpreted as a meaningful result of geometric constraint, consistent with our goal of controlled fluid manipulation. All clarifying statements and explanatory content have been incorporated into the revised manuscript (Section 2.2 in Pages 5–8, Figs. 2h–j in Page 7) and Supporting Information (Supplementary Note 2, Figs. S9–S15), marked in red.

Minor comments and suggestions:

1. In Figure 1a, the legend for the illustration will read more easily if it goes in order: cross, cusp, and square.

Response 1: We sincerely thank the respected reviewer for this thoughtful suggestion. We agree that the original labeling sequence in Figure 1a may appear slightly disordered and could affect the reader’s understanding of the schematic.

To improve clarity and logical flow, we have revised the labeling order in Figure 1a from the original sequence to “cross, cusp, and square.” This order better reflects the conceptual progression of the structural design—starting from the central bulk shape (cross or square), followed by the surrounding cusp elements that define the directional spreading behavior.

The updated figure is now presented below as **Figure R63** and has been incorporated into the revised manuscript (**Fig.1a**). This revision improves figure readability and overall visual communication. The corresponding modification has been highlighted in red font in the revised version.

Figure R63. Schematic of the cross-cusp and square-cusp microstructure.

2. In Figure 3a, when the liquid is spreading, there is a change in the meniscus that resembles the bubble or meniscus deformation in the spreading direction. Can authors comment on it and how it looks for different structures?

Response 2: We sincerely thank the reviewer for raising this critical and thought-provoking point. The observed deformation of the meniscus during spreading, particularly in Figure 3a, indeed reveals an important dynamic feature of how the three-phase contact line responds to geometric constraints on structured surfaces. This feature is not only visually distinct but also mechanistically insightful in the context of directional wetting.

(1) Clarification of the Phenomenon

We confirm that the visual appearance resembling a “bubble” is not due to air entrapment, but rather stems from a significant deformation of the advancing meniscus. This occurs as the precursor film encounters sharp tip features or sudden geometric transitions (e.g., cusp or corner), leading to a localized curvature amplification of the liquid-vapor interface. Under reflected light or specific illumination angles, such curved meniscus edges can produce high-contrast outlines or dark fringes, which may mimic the appearance of trapped voids due to optical shadowing or interference near the contact line.

(2) Comparative Visualization Across Structures

To systematically analyze this behavior, we have provided a comprehensive comparison across eight representative microstructure designs (C-mode I–IV and S-mode I–IV) in **Figure R64** and **Figure R65** of the revised Supporting Information. The side-by-side time-lapse images clearly demonstrate how the meniscus deformation emerges and evolves as the liquid spreads over each structure. Across all tested structures (C-mode I–IV and S-mode I–IV), the meniscus deformation consistently occurs during the initial stages of precursor film propagation.

These details are more clearly visualized in **Figure R66**, where we manually

extracted and overlaid the meniscus profiles (highlighted in red) at selected timepoints. A common feature lies in the sequential, anisotropic modulation of the liquid front: instead of expanding isotropically, the film deforms along preferred directions dictated by the geometry, resulting in asymmetrical front shapes, interface folding, or stepped protrusions. These deformations are not structural defects or artifacts but are reproducible manifestations of directional pinning and release. Such behavior is especially prominent when the underlying microstructure combines both directional features and lateral confinement, forming a hybrid or compound field that drives the liquid along irregular, yet controllable, paths.

Figure R64. Sequential images showing precursor film spreading on a cross-cusp structured surface.

Figure R65. Sequential images showing precursor film spreading on a square-cusp structured surface.

Figure R66. Snapshots of precursor film and meniscus morphology during spreading on cross-cusp and square-cusp structured surfaces.

(3) Supportive Image Evidence and Comparative Literature

To reinforce and contextualize our observations of meniscus deformation during precursor film spreading, we have integrated new image-based analyses and literature comparisons into the revised manuscript:

Our phenomenon resonates with the meniscus behavior observed by Chen et al. in *Nepenthes alata* peristome surfaces (Nature 2016, 532, 85–89), where stepped capillary wetting arises from ratchet-like grooves². As shown in **Figure R67(a)**, the advancing film repeatedly pins at each groove edge, creating segmented deformation that closely parallels our interface bulges in cusp-arrayed structures.

Figure R67(b) presents another prior study (ACS Appl. Mater. Interfaces, 2024, 16, 41694–41703), where stretching branch-shaped structures modulate oil droplet front deformation. The observed interface shape evolution echoes the staggered bulges and directional release seen in our current compound geometries¹.

In **Figure R67(c)**, our previous work (Nano Lett., 2023, 23, 5696–5704) showed that jaw-like tip arrays induce bulged menisci at sharp fronts during unidirectional water spreading³. The interface morphologies strongly resemble the deformation observed in the cross-tip and square-tip regions in this study.

Figure R67. Representative examples of precursor film and meniscus deformation during guided spreading on structured surfaces. a Precursor film spreading over tilted microcavities on the *Nepenthes alata* peristome surface (adapted from Chen et al.,

Nature, 2016, 532, 85–89)². **b** Meniscus morphology during unidirectional oil droplet spreading on tip-guided surfaces (adapted from ACS Appl. Mater. Interfaces, 2024, 16, 41694–41703)³. **c** Meniscus morphology evolution in water droplet spreading on jaw-like microstructures (adapted from Liu et al., Nano Lett., 2023, 23, 5696–5704)¹.

Together, these comparisons reveal a universal role of structural constraints in shaping the meniscus before directional release. Our microstructures further demonstrate programmable multi-axial spreading through predefined deformation patterns.

(4) Geometric Mechanism Interpretation

The observed meniscus deformation is fundamentally rooted in the coupling between local capillary forces and compound structural confinement. When a precursor film encounters a field of geometrically arranged cusp-like protrusions—especially those combining tip features with lateral frames—the contact line is subjected to anisotropic pinning and staggered release. This leads to transient interface bending, directional asymmetry, and occasional “step-wise” forward collapse.

Rather than representing random fluctuation, these behaviors reflect a robust geometric programming of interface dynamics. Compound microstructures provide a controllable framework in which front deformation, pinning intensity, and retraction symmetry can be tuned by altering motif orientation, connectivity, or lattice periodicity. These effects directly influence the resultant droplet shape and spreading bias, as demonstrated by our quadrant-based anisotropy metrics.

(5) Outlook and Ongoing Work

Building on these findings, future work will focus on correlating interface curvature evolution with quantitative force modeling and energy dissipation pathways. We aim to integrate high-speed top-view and side-view imaging with phase-field or volume-of-fluid (VOF) simulations to reveal real-time contact line dynamics and capillary stress distribution. This will support the rational design of programmable surfaces for applications in liquid transport, wettability control, and adaptive droplet confinement.

We have incorporated all related discussions and image-based clarifications into the revised manuscript (**Section 2.3 in Page 10, Fig. 2a**) and Supporting Information (**Supplementary Figs. S28–S29**), with added visual annotations and descriptive analysis highlighted in red. We sincerely thank the respected reviewer again for raising this valuable point, which has deepened our investigation into meniscus dynamics and further enriched the physical framework of geometry-guided spreading.

References

1. Zhang H., et al. Stretch-Controlled Branch Shape Microstructures for Switchable Unidirectional Self-Driven Spreading of Oil Droplets. *ACS Appl. Mater. Interfaces* **16**, 41694-41703 (2024).
2. Chen H.W., et al. Continuous directional water transport on the peristome surface of *Nepenthes alata*. *Nature* **532**, 85 (2016).
3. Liu Y., et al. Bionic Jaw-Like Micro One-Way Valve for Rapid and Long-Distance Water Droplet Unidirectional Spreading. *Nano Lett.* **23**, 5696-5704 (2023).
4. Li J.Q., et al. Topological liquid diode. *Sci. Adv.* **3**, eaao3530 (2017).
5. Feng S.L., et al. Three-dimensional capillary ratchet-induced liquid directional steering. *Science* **373**, 1344+ (2021).
6. Yang L., et al. Selective directional liquid transport on shoot surfaces of *Crassula muscosa*. *Science* **384**, 1344-1349 (2024).
7. Li X., et al. Bioinspired Topological Surface for Directional Oil Lubrication. *ACS Appl. Mater. Interfaces* **12**, 5113-5119 (2020).
8. Guo Y.X., et al. Bioinspired cone structures with helical micro-grooves for fast liquid transport and efficient fog collection. *J. Mater. Chem. A* **11**, 12080-12088 (2023).
9. Chen H.W., et al. Ultrafast water harvesting and transport in hierarchical microchannels. *Nat. Mater.* **17**, 935+ (2018).
10. Miao J.Q., Tsang A.C.H. Reconfigurability-Encoded Hierarchical Rectifiers for Versatile 3D Liquid Manipulation. *Adv. Sci.* **11**, 2405641 (2024).
11. Miao J.Q. Bioinspired multi-asymmetric magnetized surfaces for tailored liquid operations and 3-DOF solid transport. *Sensors and Actuators a-Physical* **368**, 115104 (2024).
12. Sun S.Q., et al. In Situ Multi-Directional Liquid Manipulation Enabled by 3D Asymmetric Fang-Structured Surface. *Adv. Mater.* **36**, 2407034 (2024).
13. Li C.Z., et al. 3D-Printed Biomimetic Shape-Memory Rectifier for Smart Directional Transport of Diverse Low-Surface-Tension Liquids and "Chip" Transfer. *Adv. Funct. Mater.*, 2507221 (2025).
14. Miao J.Q., Tsang A.C.H. Smart Directional Liquid Manipulation on Curvature-Ratchet Surfaces. *ACS Nano* **19**, 5829-5838 (2025).
15. Miao J.Q., et al. Natural Cilia and Pine Needles Combinedly Inspired Asymmetric Pillar Actuators for All-Space Liquid Transport and Self-Regulated Robotic Locomotion. *ACS Appl. Mater. Interfaces* **14**, 50296-50307 (2022).
16. Jin L., et al. Friction mechanism of DLC/MAO wear-resistant coatings with porous surface texture constructed in-situ by micro-arc oxidation. *Surf. Coat. Technol.* **473**, 130010 (2023).
17. Lin P., et al. Effect and performance analysis of different surface treatments on polymer-metal friction pairs. *Tribol. Int.* **195**, 109602 (2024).
18. Dou X., et al. Self-dispersed crumpled graphene balls in oil for friction and wear reduction. *Proceedings of the National Academy of Sciences of the United States of America* **113**, 1528-1533 (2016).

19. Hou X.B., et al. Preparation and Tribological Properties of Graphene Lubricant Additives for Low-Sulfur Fuel by Dielectric Barrier Discharge Plasma-Assisted Ball Milling. *Processes* **9**, 272 (2021).
20. Karimi S., et al. Empirical investigation of the effect of adding nanoparticles to HB-80 gas turbine oil: Evaluation of thermophysical behaviors. *Heliyon* **10**, e29759 (2024).
21. Kumar M.S., et al. Enhancement in the Friction and Wear Resistance of Low Carbon Chromium Steel and Load Carrying Capability of MIL-PRF-23699 Grade Lubricant Using h-BN Nanoadditives for Aerospace Applications. *Tribology Transactions* **66**, 882-894 (2023).
22. Liu S.C., et al. Effects of Laser Surface Texturing and Lubrication on the Vibrational and Tribological Performance of Sliding Contact. *Lubricants* **10**, 10 (2022).
23. Cao H.Y., et al. Frictional Behaviour of the Microstructural Surfaces Created by Cylindrical Grinding Processes. *Applied Sciences-Basel* **12**, 618 (2022).
24. Chen K.P., et al. Enhancing Tribological Performance of Cylinder Guide Bush through Bionic Texturing and Composite Grease. *Tribology Transactions* **68**, 558-570 (2025).
25. Ge Z.H., et al. Surface Tribological Properties Enhancement Using Multivariate Linear Regression Optimization of Surface Micro-Texture. *Coatings* **14**, 1258 (2024).
26. Cai Q.S., et al. Influence of shot peening on the microstructure and friction-wear performance of CF53 steel. *Plos One* **20**, e0317410 (2025).
27. Zhang H., et al. Effect of Ultrasonic Rolling on Surface Properties of GCr15 Spherical Joint Bearing. *Lubricants* **12**, 208 (2024).
28. Hu G.L., et al. Effect of Low-Frequency Vibration on the Tribological Properties of Thin-Walled Inconel 601 with Laser Wire Additive Manufacturing. *Adv. Eng. Mater.* **27**, 2500187 (2025).
29. Acharya S., et al. Surface mechanical attrition treatment of low modulus Ti-Nb-Ta-O alloy for orthopedic applications. *Materials Science and Engineering: C* **110**, 110729 (2020).
30. Berce J., et al. Effect of Surface Wettability on Nanoparticle Deposition during Pool Boiling on Laser-Textured Copper Surfaces. *Nanomaterials* **14**, 311 (2024).
31. Orman L.J., et al. Analysis of Enhanced Pool Boiling Heat Transfer on Laser-Textured Surfaces. *Energies* **13**, 2700 (2020).
32. Orman L.J., et al. Laser Treatment of Surfaces for Pool Boiling Heat Transfer Enhancement. *Materials* **16**, 1365 (2023).
33. Zhao H.B., et al. Microstructured Ceramic-Coated Carbon Nanotube Surfaces for High Heat Flux Pool Boiling. *Acs Applied Nano Materials* **2**, 5538-5545 (2019).
34. Sen P., et al. Pool Boiling Performance on Cu-TiO₂ Nanoparticle-Coated Copper Surfaces Prepared Through Hybrid Method. *Heat Transfer Engineering* **46**, 345-361 (2025).
35. Gupta S.K., Misra R.D. Effect of two-step electrodeposited Cu-TiO₂

- nanocomposite coating on pool boiling heat transfer performance. *J. Therm. Anal. Calorim.* **136**, 1781-1793 (2019).
36. Chun J., et al. Fast Capillary Wicking on Hierarchical Copper Nanowired Surfaces with Interconnected V-Grooves: Implications for Thermal Management. *Acs Applied Nano Materials* **4**, 5360-5371 (2021).
 37. Luo J.L., et al. Biomimetic Copper Forest Wick Enables High Thermal Conductivity Ultrathin Heat Pipe. *ACS Nano* **15**, 6614-6621 (2021).
 38. Luo J.L., et al. Biomimetic Copper Forest Structural Modification Enhances the Capillary Flow Characteristics of the Copper Mesh Wick. *Energies* **16**, 5348 (2023).
 39. Wang J., et al. Enhanced ionic wind generation by graphene for LED heat dissipation. *International Journal of Energy Research* **43**, 3746-3755 (2019).
 40. Xu C.L., et al. Enhanced Cooling of LED Filament Bulbs Using an Embedded Tri-Needle/Ring Ionic Wind Device. *Energies* **13**, 3008 (2020).
 41. Cheng S.F., et al. On Electronics Cooling Using an Electrohydrodynamic Gas Pump with Aligned Electrodes. *J. Thermophys Heat Transfer* **38**, 360-367 (2024).

Response to Comments

Reviewer #1:

The authors have revised the manuscript substantially in response to my previous comments. Considering the scope of the additional experimental results and the validity of the claims supported by them, I believe the manuscript has been significantly improved and can now be considered for publication in Nature Communications.

Response: We sincerely thank the reviewer for the positive and encouraging evaluation. We are pleased that the additional experiments and revisions have strengthened the manuscript and appreciate the reviewer's recognition of its improved quality.

Reviewer #2:

The science in this manuscript is strong, and the authors have clearly made considerable effort to address the concerns raised in the first round of review. However, I note that the current version of the manuscript has become quite long and dense. The main text contains four figures with many subpanels, and the supplementary material now spans over 50 pages with a large number of additional figures. Many of these supplementary figures are referenced in the main text, which makes it challenging to follow the narrative smoothly.

While the additional data may be scientifically valid, the sheer volume of figures and supplementary material makes the manuscript harder to read and digest. I would encourage the authors to consider ways to streamline the presentation—perhaps by consolidating subpanels, combining figures where appropriate, or ensuring that only essential supplementary figures are cited in the main text. This would improve clarity and help readers focus on the key findings without compromising the scientific rigor of the work.

Overall, the manuscript is scientifically sound, but a more concise presentation would make it more accessible and impactful.

Response: We sincerely thank the reviewer for the valuable and constructive

feedback. We fully agree that the manuscript had become relatively lengthy due to the addition of new experiments and expanded discussion to enhance scientific rigor. Following the reviewer's suggestion, we have streamlined the main text and supplementary materials by consolidating several subpanels, merging related figures, and simplifying cross-references to improve readability and clarity without affecting the completeness of the results.

Reviewer #3:

I appreciate the authors' careful and thorough revision of the manuscript. They have addressed all concerns previously raised with detailed explanations, additional analysis, and clearer presentation of results. The overall quality of the paper has improved substantially, both in terms of clarity and scientific rigor. The discussion is now more comprehensive and well-balanced, effectively situating the findings within the existing literature. I believe the manuscript is now in a publishable form and recommend it for acceptance.

Response: We sincerely thank the reviewer for the thorough assessment and positive recommendation. We are grateful for the recognition of our efforts to improve the clarity, rigor, and contextualization of the work.